# The influence of spatiality on shipping emissions, air quality and potential human exposure in Yangtze River Delta/Shanghai, China

Junlan Feng[1], Yan Zhang[1,2,3*], Shanshan Li[1], Jingbo Mao[1], Allison P. Patton[4], Yuyan Zhou[1], Weichun Ma[1,3], Cong Liu[5], Haidong Kan[5], Cheng Huang[6], Jingyu An[6], Li Li[6], Yin Shen[7], Qingyan Fu[7], Xinning Wang[7], Juan Liu[7], Shuxiao Wang[8], Dian Ding[8], Jie Cheng[9], Wangqi Ge[9], Hong Zhu[9], Katherine Walker[4]

[1] Shanghai Key Laboratory of Atmospheric Particle Pollution and Prevention (LAP3), Department of Environmental Science and Engineering, Fudan University, Shanghai 200438, China

[2] Institute of Atmospheric Sciences, Fudan University, Shanghai 200438, China

[3] Shanghai Institute of Eco-Chongming (SIEC), Shanghai 200062, China

[4] Health Effects Institute, 75 Federal Street, Suite 1400, Boston, MA 02110-1817, USA

[5] Public Health School, Fudan University, Shanghai 200032, China

[6] Shanghai Academy of Environmental Science, Shanghai 200233, China

[7] Shanghai Environmental Monitoring Center, Shanghai 200030, China

[8] State Key Joint Laboratory of Environment Simulation and Pollution Control, School of Environment, Tsinghua University, Beijing 100084, China

[9] Shanghai Urban-rural Construction and Transportation Development Research Institute, Shanghai 200032, China

*Correspondence to*: Yan Zhang (yan_zhang@fudan.edu.cn)

**Abstract**:  The Yangtze River Delta (YRD) and the megacity of Shanghai are host to one of the busiest port clusters in the world; the region also suffers from high levels of air pollution. The goal of this study was to estimate the contributions of shipping to regional emissions, air quality, and population exposure and to characterize the importance of the geographic spatiality of shipping lanes and different-type shipping-related sources for the baseline year 2015, prior to the implementation of China's Domestic Emission Control Areas (DECAs) in 2016. The WRF-CMAQ model was used to simulate the influence of coastal and inland-water shipping, in port emissions, shipping-related cargo transport on air quality and, population-weighted concentrations, a measure of human exposure. Our results showed that the impact of shipping on air quality in the YRD was attributable primarily to shipping emissions within 12 NM of shore, but emissions coming from the coastal area of 24 to 96 NM still contributed substantially to ship-related $PM_{2.5}$ concentrations in YRD. The overall contribution of ships to $PM_{2.5}$ concentration in YRD could reach to 4.62 $\mu g/m^3$ in summer when monsoon winds transport shipping emissions onshore. In Shanghai city, inland-water going ships were major contributors

(40-80%) to the shipping impact on urban air quality. Given the proximity of inland-water ships to urban populations of Shanghai, the emissions of inland-water ships contributed more to population-weighted concentrations. These research results provide scientific evidence to inform policies for controlling future shipping emissions; in particular, in the YRD region, expanding the boundary of 12 NM in China's current DECA policy to around 100 NM would include most of the shipping emissions affecting air pollutant exposure, and stricter fuel standards could be considered for the ships on inland rivers and other waterways close to residential regions.

**Key words:** Shipping, ports, emissions, source attribution, population-weighted concentration, Shanghai, Yangtze River Delta, emission control area

**1 Introduction**

With the increase of international maritime trade, shipping emissions and their impacts have attracted increased attention globally over the past decades (Capaldo et al., 1999;Cooper, 2003;Eyring et al., 2010;Sofiev et al., 2018). Shipping emits air pollutants that contribute to adverse impacts on climate, on air quality and on the health of people living near ports (Li et al., 2018;Liu et al., 2016a). Globally, about 50,000 to 90,000 cardiopulmonary diseases and lung cancer deaths were attributable to exposure to particulate matter emitted from shipping in 2010 and 2012, respectively (Corbett et al., 2007;Partanen et al., 2013;Winebrake et al., 2009), and 403,300 premature mortalities per year due to shipping are predicted in 2020 under business-as-usual (BAU) assumptions (Sofiev et al., 2018). In Europe, ozone pollution caused by international ships led to around 3.6 % of the total estimated years of life lost and 2.6 % of premature deaths in 2005 (Campling et al., 2013). In East Asia, around 14,500 to 37,500 premature deaths per year has been primarily attributed to $PM_{2.5}$ from shipping; about one third of those deaths were in the area surrounding the East China sea, with the largest impacts in mainland China (Liu et al., 2016a).

As of 2016, China was home to 7 of the top 10 container ports, and the size of those ports has been rapidly growing to serve the increased trade via international shipping (UNCTAD, 2017). The Yangtze River Delta (YRD) is one of the economic centers as well as home to the busiest port cluster, comprised of more than 15 ports, including Shanghai port, Ningbo-Zhoushan port, Zhenjiang port, Nantong port, Lianyungang port, Taizhou port, and Wenzhou port. In 2016, YRD

generated a GDP of RMB 17.72 trillion (US $2.76 trillion) – about 20 percent of China's national GDP (Preen, 2018). Shanghai megacity itself is an important economic center, accounting for about 22 % of the total GDP in YRD. Shanghai port lies at the intersection of the East China Sea and the Yangtze River and has been the largest container port in the world since 2010 (Liu et al., 2016b).

Shanghai and the YRD are also among the most densely populated regions of China. The YRD is home to 239.1 million people; Shanghai is one of the largest cities and houses about 12.1 % of the total population of the YRD (Bright et al., 2016).

This region has suffered from severe air pollution over the past decade due to the anthropogenic emissions from multiple sources. In December 2013, for example, YRD experienced a haze episode, in which the maximal observed $PM_{2.5}$ concentration in YRD exceeded 590 μg/m$^3$ (Sun et al., 2016). As severe air pollution episodes have continued and ports have grown, the shipping sector, a subset of transportation pollution sources, has received more attention.

The high ship traffic density in Shanghai and YRD has led to high emissions of shipping-related air pollutants in this region (Fan et al., 2016). Shipping-related sources of air pollution in Shanghai comprise coastal ships, inland-water ships, container-cargo trucks, and port terminal equipment. Because some of these emissions sources are also close to densely populated areas, in particular those from ships traveling in inland waterways and from container trucks transporting cargo in and around the city, there is greater potential for higher population exposures to ship-related air pollution.

The International Maritime Organization (IMO) regulates emissions of marine pollution on a global scale. Current rules limit fuel sulfur content (FSC) to 3.5 % globally and will lower this limit to 0.5 % in 2020. The IMO has also designated several regional Emission Control Areas (ECAs) to benefit the atmospheric environment and human health in port and coastal communities that establish more stringent emissions limits up to 200 NM from the coast in the Baltic Sea ($SO_X$), North Sea ($SO_X$), North America ($SO_X$, $NO_X$, and PM), and the United States Caribbean Sea area ($SO_X$, $NO_X$, and PM) (Viana et al., 2015). Fuel sulfur content is limited to 0.1 % in the ECAs.

China does not have an ECA designated by the IMO, but in December 2015 it designated three Domestic Emission Control Areas (DECAs) that operate in a similar manner. These DECAs

limited fuel sulfur content to 0.5 % for ocean-going vessels (OGV) in 3 regions: YRD, Pearl River Delta (PRD) and Bohai Sea. The DECA implementation timeline specified that qualified ports are encouraged to be in compliance since April 1, 2016, and all ships at berth in 11 core ports within these regions would be in compliance by January 1, 2017 and all ocean-going vessels (OGV) or coastal vessels within 12 NM of the shoreline would be in compliance by January 1, 2019. These areas would also be in compliance with the IMO requirements for fuel sulfur content. A study reported that the average reduction of $PM_{2.5}$ and $SO_2$ mass concentrations over land in the PRD due to the DECA policy were 2.7% and 9.54% (Liu et al., 2018a). China is currently considering additional DECA restrictions for the period beyond 2019. Starting on October 1, 2018, three months earlier than the original plan, the Shanghai Maritime Safety Administration (MSA) has enforced the DECA policy limiting fuel sulfur content to 0.5 % for ocean-going vessels and domestic coastal vessels in Shanghai port. However, the DECA policies for fuel sulfur content currently make no distinction between coastal ships that enter inland water areas and other ships. Ships like those in Shanghai and the YRD that enter inland waterways bring emissions sources closer to population centers resulting in a greater potential for exposure and health impacts.

Shipping emission inventories for the YRD, PRD, and Bohai-Rim area and their major ports indicate that shipping is an important pollution source surrounding port regions (Chen et al., 2016;Fan et al., 2016;Li et al., 2016;Yau et al., 2012). Several studies have investigated the contribution of shipping emissions to ambient air quality using different methods. Zhao et al. (2013) analyzed aerosol samples in Shanghai Port and reported that ship traffic contributed 0.63 $\mu g/m^3$ to 3.58 $\mu g/m^3$ (or 4.2 % to 12.8 %) of the total $PM_{2.5}$ in Shanghai Port. Primary ship-emitted particles measured by an aerosol time-of-flight mass spectrometer were typically 1.0 to 10.0 % of the measured particle number concentration, with the contribution rising to as high as 50.0 % in spring and summer (Liu et al. 2016b). In Guangzhou and Zhuhai, shipping emissions were among the top contributors to $PM_{2.5}$ and accounted for greater than 17% of $PM_{2.5}$ mass concentrations (Tao et al. 2016). Using WRF-CMAQ, Chen et al. (2017b) found that the contribution of shipping emissions to the $PM_{2.5}$ mass concentrations in Qingdao was the highest in summer (13.1%) and the lowest in winter (1.5 %). Chen et al. (2019) reported that ship traffic sources could contribute 4.0 % of annual $PM_{2.5}$ mass concentrations over the land area in YRD and the maximum could reach 35.0% in port region in 2014.

In China, a few studies reported the contribution to air pollution from shipping in different offshore coastal areas or different-type ship-related sources. For example, Mao et al. (2017) estimated primary emissions from OGVs at different boundaries in the PRD region, and concluded that further expansion of emission control area to 100 NM would provide even greater benefits. However, the impacts of shipping emissions at varying distances from shore on air quality and potential human exposure, which are important when considering ECA policy, have not been rigorously studied. Mao and Rutherford (2018) studied $NO_x$ emissions from three categories of merchant vessels—OGVs, coastal vessels (CVs) and river vessels (RVs) in China's coastal region. But less attention was paid to the impacts of inland waterway traffic and port-related sources like container-cargo trucks and terminal port equipment on air quality and potential human exposure.

To fill this gap, the overall goal of this study was to characterize the spatial distribution of shipping-related emissions and their impacts on air quality and human exposure in the YRD and Shanghai for the baseline year 2015, prior to the implementation of China's DECAs in 2016. We modeled shipping emissions in different offshore areas in the YRD region and emissions from different-type ship-related sources in Shanghai city for each month of the year. To identify which offshore areas in the YRD region and which ship-related sources in Shanghai contributed the most ambient air pollution, and human population exposure, we modeled the impacts of shipping emissions in different offshore areas (within 12 NM including inland waters, 12-24 NM, 24-48 NM, 48-96 NM, and 96-200 NM) in the YRD region as well as coastal ships, inland-water ships, and container-cargo trucks and port terminal equipment in and near the port areas under the jurisdiction of Shanghai MSA in two representative months (January and June). The results of this study could be informative to the consideration of the distance of regulated emissions in the design of future emissions control areas for shipping in YRD, or regulations on the sulfur content of fuels for different-type ship-related sources in Shanghai.

**2 Methodology**

In this study, we first established a shipping emission inventory based on highly-resolved automatic identification system (AIS) data in 2015. Then, we used WRF-CMAQ model to evaluate the impacts on air quality from shipping emissions in different offshore coastal areas (within 12 NM including inland waters, 12-24 NM, 24-48 NM, 48-96 NM, and 96-200 NM) in the YRD region. We referred to ICCT's working paper (Mao et al., 2017) to choose the distance bins

between 12 NM (the boundary of current China's DECA) and 200 NM (the boundary of ECA designated by IMO) in this study. The model domains were shown in Figure S1. Simulations were also conducted to estimate the influence of different-type shipping-related sources (coastal ships, inland-water ships, container-cargo trucks, and port terminal equipment) on air quality in Shanghai. Finally, population-weighted $PM_{2.5}$ concentrations attributable to shipping sources were calculated.

### 2.1 Study area and period

Figure 1 shows the geographic area and population density for the YRD and Shanghai, the location of 16 core cities of the YRD region, and 16 administrative districts within Shanghai city. The coastal cities in the YRD are Nantong, Shanghai, Jiaxing, Ningbo, Taichou, and Zhoushan.

The simulation network was developed for four domains at resolutions of 81 km $\times$ 81 km, 27 km $\times$ 27 km, 9 km $\times$ 9 km, and 1 km $\times$ 1 km, respectively (Fig. S1). Domain 1 covers the whole China. Nested domains 2, 3, and 4 cover a large part of East-China (2), the YRD region (3, including Jiangsu, Zhejiang, and Shanghai), and Shanghai with a finer resolution (4), respectively. The geographic scope for the YRD study area extended from 116.5 °E to 127 °E and 27 °N to 35 °N and included an offshore distance of approximately 200 NM. The Shanghai study area included from 120.5 °E to 122.3 °E and from 30.5 °N to 32 °N, where the water is within the jurisdiction of Shanghai MSA.

Two representative months in the year 2015, January and June, were selected to compare the seasonal effects. Higher shipping impacts were expected in June because prevailing winds from the summer monsoon are directed from the ocean to the shore. January was chosen as a contrasting period with prevailing winds away from shore.

### 2.2 Emission inventories

### 2.2.1 Ship-related emission inventories

In this study, emission inventories were constructed based primarily on automatic identification system (AIS) data for shipping traffic activity in China, YRD, and Shanghai geographic domains. Due to limitation of the national-scale data source, the AIS data in this study only covered the representative months of January and June 2015, while the YRD-scale AIS data covered 2015 full year. AIS data includes international ships, coastal ships, and inland-water ships, but some river ships could be not covered by AIS data. Emissions from ships entering the

geographic domains for YRD or Shanghai were calculated using the AIS-based model developed by Fan et al. (Fan et al., 2016), and monthly shipping emissions for January and June were used in the air quality model to capture the seasonal variation to expect more accurately than annual shipping emissions with no monthly variations. For Shanghai, estimates of emissions from those ships without AIS devices were supplemented by using 2015 vessel call data provided by Shanghai MSA and Shanghai Municipal MSA. The detailed method, assumptions and sources are provided in section S.1 of the supporting information. The actual speeds and operation times of the ships involved in the calculation can be obtained from AIS data with high accuracy, while the installed power of the main engine (ME), auxiliary engine (AE), and auxiliary boiler (AB) and the maximum speed of ships necessary to complete the estimates were obtained from Lloyd's register (now IHS-Fairplay) (Lloyds, 2015) and the China Classification Society (CCS) database. Assumptions regarding the fuel types, sulfur contents and engine types, and sources of emission factors, low load adjustment multipliers, and control factors are provided in section S.2 of the supporting information.

Within the Shanghai port domain, separate emissions inventories were developed to estimate the relative air quality impacts of coastal and inland-water and of ship-related container-cargo trucks transport and port terminal equipment (cranes, forklifts, and trucks used for internal transport). Many coastal ships operate in both the outer port and in the inner river region of Shanghai Port, which includes the Yangtze River, Huangpu River and other rivers in Shanghai. Consequently, a geographic boundary was used to divide the shipping emissions inventory based on AIS data into coastal and inland sources (see Figure 3c in which the black line denotes a division between coastal and inland shipping contributions to emissions).

Emissions from container-cargo trucks were estimated using International Vehicle Emission (IVE) model (Wang et al., 2008). The vehicular activity data was provided by the Shanghai Traffic Department. The emissions from port terminal equipment including the trucks in port were calculated based on fuel consumption for each part of the port. Given their smaller emissions relative to shipping and other non-port sources, emissions from container-cargo trucks and terminal equipment were combined and gridded at a resolution of 1 km $\times$ 1 km.

**2.2.2 Non-shipping emission inventories**

National and local YRD emission inventories were used for emissions from all other sources

(non-shipping). For the national scale domain, we used a 2015 national emission database at a 27 km $\times$ 27 km resolution that included 5 pollutants ($PM_{10}$, $PM_{2.5}$, $SO_2$, $NO_x$ and VOCs) and 14 source types (see Table S1 for details) ( Zhao et al., 2018). Since the national emission inventory database lacked data on CO and $NH_3$ emissions, which are compulsory inputs for CMAQ model, supplemental emission data on these pollutants in 2015 were obtained from the International Institute for Applied Systems Analysis (IIASA) database (at a $0.5\,°\times 0.5\,°$ resolution) (Stohl et al., 2015). In case of the large uncertainty leaded by merging data from two datasets, the ratio of CO to VOC was checked in this study. CO and VOC emissions both result from incomplete combustion of fuel and are likely to be related (von Schneidemesser et al., 2010; Wang et al., 2014). The ratio of CO to VOC was 7.7 in the IIASA inventory and 7.5 in the final combined inventory. Thus, the CO/VOC shares in these two inventories were very close and the use of the final combined inventory is acceptable. The Shanghai Academy of Environmental Sciences (SAES) provided the local YRD land-based emission inventory at a 4 km $\times$ 4 km resolution; it included 8 source types and 7 pollutants ($PM_{10}$, $PM_{2.5}$, $SO_2$, $NO_x$, CO, VOCs and $NH_3$). Details are provided in Table S2. National and local emission data were allocated to simulation grids by spatial interpolation in ArcGIS 10.2 (ESRI, 2013).

### 2.3 WRF-CMAQ model setup

The models used in this study were the Weather Research and Forecasting Model (WRF) version 3.3 and the Community Multiscale Air Quality (CMAQ) model version 4.6. The selected simulation periods were 1 January to 31 January and 1 June to 28 June, with 72 hours of spin-up time for each run. The initial and boundary conditions for meteorology were generated from the National Centers for Environmental Prediction (NCEP) Final Analysis (FNL) (NCEP, 2000) with resolution at $1\,°\times 1\,°$ at six hour time intervals. Vertically, 27 sigma layers were set for the WRF simulation, and the results were then converted to the 24 layers required by CMAQ (version 4.6) using the MICP (Meteorology-Chemistry Interface Processor). CMAQ was configured to use the Carbon Bond mechanism (CB05) for gas-phase chemistry and the AERO4 aerosol module (Liu et al., 2016b).

### 2.4 Simulations of source contribution to air quality

Individual source contributions to gridded ambient concentrations of air pollution were estimated as the difference between the concentrations simulated with all sources included and those with the individual source excluded. For the YRD region (domain 3), the simulation was conducted for ships within different boundaries from shore (12 NM, 12-24 NM, 24-48 NM, 48-96 NM and 96-200 NM). For the city of Shanghai, simulations were conducted for all ship-related sources in the water area under the jurisdiction of Shanghai MSA (within approximately 12 NM of shore), coastal and inland-water shipping (as defined geographically above), and container-cargo transport and port terminal equipment (combined). Details of each simulation can be found in Table S3.

**2.5 Model evaluation**

Performance of the models was spatially evaluated by comparison with monthly-average observations at monitoring stations (Fig. S2). Generally, the simulated results showed trends consistent with the observations, with increased concentrations of $SO_2$ and $PM_{2.5}$ along the Yangtze River and in the urban areas. Also, daily-average observations from 53 monitoring stations in 16 core YRD cities were compared with daily-average simulated ambient $SO_2$ and $PM_{2.5}$ concentrations. Normalized Mean Bias (NMB), Normalized Mean Error (NME), Root Mean-square Error (RMSE), and Pearson's correlation coefficient ($r$) were used to qualify the degree of deviation between the observed data and modeling results (Eder and Yu, 2007). Detail equations of the above statistical metrics are shown in section S.3. For each of the cities, the statistical metrics were calculated based on the average observed data and simulated results of the monitoring stations in the city, as shown in Table 1. For most cities, $SO_2$ and $PM_{2.5}$ concentrations were underestimated to varying degrees, which NMB was in the range of -36% to -18% and -34% to 8%, respectively. The deviations between the simulation results and the monitoring data were mainly due to the uncertainties of emission inventories and some deficiencies of meteorological and air quality models. However, there were also uncertainties associated with the measurements themselves and the comparison of grid-based predictions to measurements at point locations. The daily variability of simulated and observed $SO_2$ and $PM_{2.5}$ concentrations in four representative cities (two coastal cities and two inland cites) was displayed in Fig. S3, which indicates that the temporal variability of the simulated data was consistent with the observed data and the air quality

model could capture the pollution peak in most times.

**2.6 Population-weighted PM$_{2.5}$ concentration**

We estimated population-weighted PM$_{2.5}$ concentrations for the 16 cities of the YRD region and the 16 districts with Shanghai city. The population-weighted PM$_{2.5}$ concentration of the given grid cell $i$ is calculated based on Eq. (1) (Prasannavenkatesh et al., 2015):

$$\text{Population-weighted PM}_{2.5} \text{ concentration} = \sum_{i=1}^{n}\left(PM_i \times \frac{P_i}{\sum_{i=1}^{n} P_i}\right) \qquad (1)$$

where, $PM_i$ is defined as the PM$_{2.5}$ concentration in the $i$th grid cell, $P_i$ is the population in the $i$th grid value of, and $n$ is the number of grid cells in the selected geographical area, for example city or region.

Soares et al. (2014) built a refined model for evaluating population exposures to ambient air pollution in different microenvironment. In this study, in the absence of detailed individual exposure estimates, population-weighted PM$_{2.5}$ concentrations are a better approximation of potential human exposure because they give proportionately greater weight to concentrations in areas where most people live. Population-weighted exposures have been adopted as the basis for estimating the burden of disease from air pollution in the Global Burden of Disease project run by the Institute for Health Metrics and Evaluation (Cohen et al. 2017). IHME's exposure methodology is also now used by the World Health Organization.

## 3 Results and Discussion

### 3.1 Characteristics of shipping emissions

We estimated $7.2 \times 10^5$ tons of annual SO$_2$ emissions from ships in China in 2015 taking January and June as the two reference months (see Section 2.2.1 for data description and Fig. 2a for the spatial pattern). Below, we discuss the quantity and other characteristics of primary emissions from ships in different offshore coastal areas in YRD regions (Section 3.1.1) and from different ship-related sources in Shanghai (3.1.2).

### 3.1.1 Shipping emissions in YRD region

Based on the whole year 2015 AIS data, the annual emissions of SO$_2$, NO$_X$, PM$_{2.5}$, and VOC$_s$ from shipping sectors in YRD region were estimated at $2.2 \times 10^5$ tons (one third of the value for

China), $4.7 \times 10^5$ tons, $2.7 \times 10^4$ tons, and $1.2 \times 10^4$ tons, respectively, which accounted for 7.4%, 11.7%, 1.3%, and 0.3% of the total emissions from all sources in the YRD in 2015. The emission estimates of $SO_2$ and $NO_X$ were close to Fu et al.'s estimates for 2013 year, but estimates of $SO_2$, NOx and $PM_{2.5}$ were slightly lower than Chen et al.'s estimates for 2014 year due to the different temporal or spatial statistical scope (Chen et al., 2019; Fu et al., 2017). However, the proportion of ship $SO_2$ emissions of YRD region accounting for the whole China in this study is consistent with the 33% to 37% in the other studies (Chen et al., 2017a; Chen et al., 2019; Liu et al., 2018b; Lv et al., 2018)

More than 60% of annual emissions of $SO_2$ from ships in the YRD occurred inland or within 12 NM of shore, where 75.0% of the $NO_x$ emissions and 48.4% of the $PM_{2.5}$ emissions from ships occurred (Table 2). Similar results were obtained in other studies (Li et al., 2018; Liu et al., 2018a). Our estimate of $1.3 \times 10^5$ tons of annual $SO_2$ emissions emitted by ships on inland waters or within 12 NM of shore was 47% higher than Liu et al.'s estimate of $8.83 \times 10^4$ tons. However, our estimate of average emission intensity of $SO_2$ within 12 NM of shore in the YRD was 0.66 ton/yr/km$^2$, much lower than Liu et al.'s estimate of 4.04 ton/yr/km$^2$. One explanation for the different results may be that the YRD has a longer coastline than the PRD which leads to larger total emissions but to lower intensity. Emissions occurring within 24-48 NM and 48-96 NM from shore were not negligible; annual $SO_2$ emissions in these two areas accounted for 11.4% and 14.9% of the total shipping emissions in the YRD, respectively. The spatial pattern of annual $SO_2$ emissions from ships varied in different offshore coastal areas in the YRD (Fig. 2b). $SO_2$ emissions were also high at the intersection of Yangtze River and Huangpu River, between 24 and 48 NM from shore and in the north-south shipping lanes between 48 and 96 NM from shore.

### 3.1.2 Emissions from different ship-related sources in Shanghai

The annual emissions of $SO_2$, $NO_x$, $PM_{2.5}$, and $VOC_s$ from all ship-related sources within the administrative water area of Shanghai in 2015 were $4.9 \times 10^4$ tons, $1.4 \times 10^5$ tons, $6.5 \times 10^3$ tons, and $4.7 \times 10^3$ tons, respectively. The breakdown of emissions from ship-related sources in Shanghai are shown in Table 3. The emissions of $SO_2$, $NO_X$, $PM_{2.5}$, and $VOC_s$ from inland-water ships and coastal ships accounted for the majority of primary emissions from all shipping related sources in Shanghai port, ranging from 72% for VOCs to about 99% for $SO_2$. They comprised about 17.4%

of $SO_2$, 24.5% of $NO_x$, 5.2% of $PM_{2.5}$ and 0.6% of $VOC_s$ emissions from all pollution sources in Shanghai. The shipping emissions in Shanghai port were estimated to account for 23% of $SO_2$, 26% of $NO_x$, 23% of $PM_{2.5}$, and 28% of $VOC_s$ from total shipping emissions in YRD.

Emissions estimates from this study fall within the range of estimates from other studies (Fu et al., 2012; Fu et al., 2017). On the basis of shipping visa data, Fu et al. (2012) determined that the total amounts of $SO_2$, $NO_X$, and $PM_{2.5}$ in the vicinity of Shanghai port in 2010 were $3.5 \times 10^4$ ton/yr, $4.7 \times 10^4$ ton/yr, and $3.7 \times 10^3$ ton/yr, respectively, substantially lower than estimates in this study. Using AIS data, Fu et al. (2017) reported $5 \times 10^4$ tons of $SO_2$ and $7 \times 10^4$ tons of NOx from shipping in Shanghai port in 2013, close or a bit lower than the results in this study.

Within Shanghai, following the geographical division, inland-water ships were the most important ship-related source of emissions, accounting for 67% of $SO_2$, 66% of $NO_x$, 62% of $PM_{2.5}$ and 57% of VOC emissions from all ship-related sources in Shanghai (Table 2). Emissions of $SO_2$, $NO_X$, $PM_{2.5}$ and VOCs from cargo trucks and port terminal equipment comprised a smaller percentage of emissions from all shipping related sources and particularly from all pollution sources so were therefore combined into one category in model simulation.

The spatial patterns of annual emissions from ship-related sources in Shanghai are shown using $SO_2$ as an example in Fig. 2c and Fig. 2d. $SO_2$ emissions from coastal ships were more prominent on the east-west shipping lanes and the vicinity of Yangshan port (Fig. 2c) while $SO_2$ emissions from inland water-going ships were significant concentrated along the Yangtze River and the Huangpu River, which run through the center of Shanghai.

### 3.2 The impact of shipping emissions on air quality

### 3.2.1 Contribution to ambient concentrations of $SO_2$ and $PM_{2.5}$ from all ships in YRD

On average, ships contributed 0.55 μg/m$^3$ in January (Fig. 3a) and 0.73 μg/m$^3$ in June (Fig. 3c) to the land ambient $SO_2$. The contribution of shipping emissions to the ambient monthly-average $SO_2$ concentration was higher in June 2015 than in January 2015 in the YRD region. The contribution from ships to land ambient $SO_2$ concentration peaked at 6.0 μg/m$^3$ (24.3% of ambient $SO_2$) in January and 8.84 μg/m$^3$ (69.7% of ambient $SO_2$ from all pollution sources) in June.

On average, ships contributed 0.36 μg/m$^3$ in January (Fig. 3b) and 0.75 μg/m$^3$ in June (Fig.

3d) to the ambient $PM_{2.5}$ concentrations across the YRD. Similarly, the contribution of shipping emissions to ambient monthly-average $PM_{2.5}$ concentrations was higher in June 2015 than in January 2015 in the YRD region. The contribution from ships to ambient $PM_{2.5}$ concentration peaked at 1.84 μg/m$^3$ (2.2% of the total ambient $PM_{2.5}$ concentration from all pollution sources) in January and 4.62 μg/m$^3$ (18.9 % of total ambient $PM_{2.5}$) in June. The highest shipping contributions to $PM_{2.5}$ were located near the Shanghai port.

The differences between January and June contributions of shipping to air quality mainly reflect differences in meteorology. The summer monsoon winds in June flow from the sea toward and, transporting shipping emissions inland in June whereas the winter monsoon winds in January transport shipping emissions out to sea. Differences in shipping emissions did not explain the different results for January and June. Monthly shipping emissions in YRD were $1.9 \times 10^4$ tons of $SO_2$ and $2.3 \times 10^3$ tons of $PM_{2.5}$ in January and $1.8 \times 10^4$ tons of $SO_2$ and $2.3 \times 10^3$ tons of $PM_{2.5}$ in June.

### 3.2.2 The influence of different offshore coastal areas in YRD on air quality

Shipping emissions on inland waters or within 12 NM of shore accounted for 30% to 85% of the total air quality impacts of ships within 200 NM of shore in January and June 2015, respectively (Fig. 4). These results are similar to those of Lv et al. (2018) who reported that shipping emissions within 12 NM of shore contributed 30% to 90% of the $PM_{2.5}$ induced by emissions within 200 NM. On average, ships contributed 0.24 μg/m$^3$ to the ambient $PM_{2.5}$ in January (Fig.4a) and 0.56 μg/m$^3$ to ambient $PM_{2.5}$ concentrations in June (Fig.4f). Peak contributions were 1.62 μg/m$^3$ $PM_{2.5}$ in January and 4.02 μg/m$^3$ $PM_{2.5}$ in June, respectively.

The average and peak contributions from the shipping emissions in specific offshore coastal areas to the ambient $SO_2$ and $PM_{2.5}$ concentrations on shore for the two months are listed in Table S4. Shipping emissions beyond 12 NM had a much smaller impact on ambient $SO_2$, which average contributions were below 0.01 μg/m$^3$ and peak contributions were below 0.06 μg/m$^3$ (Table S4).

Shipping emissions at distances of 12-24 NM, 24-48 NM and 48-96 NM from shore contributed on average 0.01-0.07 μg/m$^3$ to the ambient $PM_{2.5}$ concentrations. Peak contributions of shipping emissions from areas beyond 12 NM ranged from 0.05 μg/m$^3$ (12-24 NM) to 0.14 μg/m$^3$ (48-96 NM) in January (Fig. 4b-d); the peak influence was higher in June and ranged from 0.2

$\mu g/m^3$ (12-24 NM) to 0.34 $\mu g/m^3$ (24-48 NM) (Fig. 4g-i). In the YRD region, shipping emissions on inland waters or within 12 NM of shore had larger contributions to ambient $PM_{2.5}$ than did more distant ships, but the busy north-south shipping lanes in the distant region from shore also impacted ambient $PM_{2.5}$ concentrations. Shipping emissions from 96 to 200 NM from shore had little impact on air quality over land and contributed less than 0.05 $\mu g/m^3$ (or 3% of the ship-related contribution) to the ambient land $PM_{2.5}$ (Fig. 4e and Fig. 4i).

The cumulative contributions to ambient $SO_2$ concentrations in the 16 core YRD cities from ships at different distances from shore in January and June 2015 differed from $PM_{2.5}$ results (Fig. 5). In both January (Fig. 5a) and June (Fig. 5c), shipping emissions within 12 NM accounted for at least 78% of the ship-related contribution to ambient $SO_2$ concentrations in these cities. Shipping emissions beyond 12 NM had limited contribution to $SO_2$ concentrations in 16 core YRD cities, implying that the boundary of 12 NM might be suitable for regulating $SO_2$ emissions. This could also be proved by Schembari et al., (2012), who reported that statistically significant reductions of $SO_2$ levels (66% to 75%) were found in 3 out of the 4 European harbours, 5 months after the implementation of the EU directive 2005/33/EC that requires all ships at berth or anchorage in European harbours use fuels with a sulfur content of less than 0.1% from January 2010. The quicker chemical reaction and shorter lifetime of $SO_2$ may explain why ships further out than 12 NM had much smaller impact on land ambient $SO_2$ concentrations (Collins et al., 2009; Krotkov et al., 2016). $SO_2$ reacts under tropospheric conditions via both gas-phase processes (with OH) and aqueous-phase processes (with $O_3$ or $H_2O_2$) to form sulfate aerosols, and is also removed physically via dry and wet deposition (Seinfeld and Pandis, 2006). The sulfur deposition due to shipping emissions is mainly contributed by the dry depositions (Chen et al.,2019). In the Planet boundary layer (PBL), $SO_2$ has short lifetimes (less than 1 day during the warm season) and are concentrated near their emission sources (Krotkov et al., 2016).

In contrast to $SO_2$, the cumulative contributions to $PM_{2.5}$ in the 16 core YRD cities from ships at different distances from shore showed greater differences in January and June 2015. In January, the relative contributions of ships inland or within 12 NM of shore to ship-related $PM_{2.5}$ concentrations ranged from 78.7% in Zhoushan, which were mostly influenced by the closest shipping emissions, to 26.3% in Yangzhou (Fig. 5b). In June, the relative contributions of ships inland or within 12 NM of shore to all $PM_{2.5}$ emissions from ships ranged from 85.2% in Nanjing

to 54.6% in Taizhou (Fig. 5d). Therefore, in both months, shipping emissions within 12 NM were a major contributor to ship-related $PM_{2.5}$ concentrations in most of core YRD cities. Although busy north-south shipping lanes 24-96 NM from shore contributed little $SO_2$ concentrations to YRD cities, shipping emissions from this area contributed 12% to 39% of ship-related $PM_{2.5}$ concentrations in YRD cities. Of $PM_{2.5}$ in YRD cities contributed by ships within 200 NM of shore, 97% is accounted for by shipping emissions within 96 NM of shore. The results of these YRD analyses suggest that although ambient ship-related $SO_2$ concentrations were mainly affected by shipping inland or within 12 NM, expanding China's current DECA to around 100 NM or more would reduce the majority of the impacts of shipping on regional $PM_{2.5}$ pollution. It also implies that the future ECA policy should consider multiple air pollutants including the primary and secondary pollutants synchronically.

### 3.2.3 The influence of different ship-related sources in Shanghai port on air quality

The impact of port-scale shipping-related sources on the air quality in Shanghai was significant, and the dominant sources of shipping-related emissions (i.e., coastal ships, inland-water ships, and other shipping-related sources) varied depending on the season and their locations relative to cities (Figure 6). Inland-water ships had a larger influence on areas within Shanghai near the Yangtze River and Huangpu River. Inland-water ships contributed on average 0.24 μg/m$^3$ in January (Fig.6a) and 0.37 μg/m$^3$ in June (Fig.6d) to ambient $PM_{2.5}$, and accounted for 40% to 80% of all $PM_{2.5}$ from ship-related sources. The inland-water ships had their large influence in areas near the cross section of Yangtze River and Huangpu River, where their contributions to ambient $PM_{2.5}$ peaked at 1.87 μg/m$^3$ in January and 2.67 μg/m$^3$ in June (Fig.6a and Fig. 6d). Coastal ships contributed on average 0.02 μg/m$^3$ in January and 0.30 μg/m$^3$ in June to ambient land $PM_{2.5}$ concentrations. Peak contributions of coastal ships to ambient $PM_{2.5}$ were 0.1 μg/m$^3$ in January (Fig.6b) and 0.71 μg/m$^3$ in June (Fig.6e). The impact of coastal ships was much smaller in January than in June due to meteorological reasons described earlier. Container-cargo trucks and port terminal equipment contributed on average 0.15 μg/m$^3$ in January (Fig.6c) and 0.12 μg/m$^3$ in June (Fig.6f) to ambient $PM_{2.5}$ concentrations, and accounted for 10 to 45% of $PM_{2.5}$ from shipping-related sources. Peak contributions of container-cargo trucks and port terminal

equipment were 2.14 μg/m$^3$ in January and 1.40 μg/m$^3$ in June. The slightly larger contribution of container-cargo trucks and terminal equipment to PM$_{2.5}$ concentrations was mainly because the lower wind speed in winter hindered the dispersion of pollutants. Although the contributions of container-cargo trucks and port terminal equipment to ambient PM$_{2.5}$ were generally lower than the contributions of ships, these other shipping-related sources were still important in both winter and summer due to their impact on air quality near the Shanghai city center.

### 3.3 Population-weighted PM$_{2.5}$ concentrations

### 3.3.1 Influence of different offshore coastal areas in YRD on population-weighted PM$_{2.5}$

Population-weighted PM$_{2.5}$ concentrations in the YRD from shipping sources were larger in June (0.4 μg/m$^3$ to 2.6 μg/m$^3$ in June; Fig. 7d) than in January (0.1 μg/m$^3$ to 1.2 μg/m$^3$; Fig. 7b). This is in contrast to population-weighted PM$_{2.5}$ concentrations from all pollution sources, which were higher in January (33.1 μg/m$^3$ to 80.2 μg/m$^3$; Fig. 7a) than in June (9.5 μg/m$^3$ to 48.4 μg/m$^3$; Fig. 7c). Thus, population-weighted PM$_{2.5}$ concentrations from shipping sources accounted for 0.9% to 15.5% of the population-weighted PM$_{2.5}$ concentrations from all pollution sources in June, larger than the contributions of 0.2% to 1.6% in January, which was attribute to higher shipping-related population-weighted PM$_{2.5}$ concentrations in June and higher population-weighted PM$_{2.5}$ concentrations from all pollution sources in January. Of the 16 core YRD cities, the highest ship-related population-weighted PM$_{2.5}$ concentrations were found for Shanghai in June (2.6 μg/m$^3$), 1.5 times higher than the second-highest city Nantong (1.7 μg/m$^3$). The six cities in the YRD with the largest contributions of PM$_{2.5}$ from shipping sources were all coastal cities, which suggests as expected that people living in coastal regions would have higher exposures to air pollution from shipping-related sources than people living in farther inland, especially during the summer monsoon.

Taking the population-weighted PM$_{2.5}$ concentrations from all shipping sources within 200NM as the base, the shipping, both in inland waters and within 12NM of shore, was a major contributor to population-weighted PM$_{2.5}$ concentrations in 16 YRD cities; they accounted for 52.9% to 82.7% (Fig. 7e). The Population-weighted PM$_{2.5}$ concentrations from shipping within 12-24NM from shore were much smaller, accounting for 2.5% to 6.6%. But shipping emissions in the

area24-48 NM accounted for 6.8% to 11.5% and ships 48-96 NM from shore accounted for 6.3% to 31.6%. These contributions in greater distance were larger than the contribution from ships in 12-24 NM from shore, probably because the busier shipping lanes fall within the more remote areas like 24-48 NM from shore. Therefore, although shipping inland and within 12 NM of shore was the dominant contributor to potential population exposure to $PM_{2.5}$, ships as far as 24-96 NM could also be important.

### 3.3.2 The influence of different ship-related sources in Shanghai port on potential exposure

Of the shipping-related sources in Shanghai, inland-water ships were the largest contributors to both $PM_{2.5}$ and population-weighted $PM_{2.5}$ (Fig. 8b). The population-weighted $PM_{2.5}$ in January was 0.38 μg/m$^3$ from inland-water ships (Fig. 8a). In June, the population-weighted $PM_{2.5}$ from inland-water ships contributed reached 0.57 μg/m$^3$ because the region near the Huangpu River and Yangtze River had a high population where inland-water ships contributed high levels of $PM_{2.5}$ (Fig. 8b). In contrast, coastal ships contributed 0.27 μg/m$^3$ and container-cargo trucks and port terminal equipment contributed only 0.14 μg/m$^3$ to population-weighted $PM_{2.5}$ in June. Population-weighted $PM_{2.5}$ from shipping sectors in January were lower than those in June, while population-weighted $PM_{2.5}$ from container-cargo trucks and port terminal equipment was slightly higher. In both June and January, population-weighted $PM_{2.5}$ concentrations from ship-related sources were larger than the average $PM_{2.5}$ concentrations from ship-related sources because the population was denser in the areas most highly influenced by shipping-related sources (Fig. 8a and 8b). The difference between average $PM_{2.5}$ concentration and population-weighted $PM_{2.5}$ concentration was largest for inland-water ships, which contributed two times more population-weighted $PM_{2.5}$ concentration than the average $PM_{2.5}$ concentration.

Population-weighted $PM_{2.5}$ concentrations were not evenly distributed among the 16 administrative districts in Shanghai. The population-weighted $PM_{2.5}$ from all pollution sources ranged from 44.8 μg/m$^3$ to 124.5 μg/m$^3$ in January (Fig. 8c) and 23.4 μg/m$^3$ to 67.2 μg/m$^3$ in June (and Fig. 8g). Heavy motor vehicle traffic probably contributed to higher population-weighted $PM_{2.5}$ in the city center (Huangpu, Jingan and Hongkou).

Areas in the city center had high population-weighted $PM_{2.5}$ from inland-water ships because

of the combination of dense population and location close to Huangpu River (Fig. 8d and Fig. 8h). Among them, Baoshan and Yangpu had the highest population-weighted $PM_{2.5}$ concentrations from inland-water ships (both around 1.31 μg/m$^3$) in June. Besides, in June, population-weighted $PM_{2.5}$ from coastal ships ranged from 0.17 μg/m$^3$ to 0.40 μg/m$^3$, and the coastal district (Fengxian) suffered the largest impacts. Transport of emissions by the summer monsoon caused impacts on population-weighted $PM_{2.5}$ not only in coastal districts but also in the highly populated city center. As for population-weighted $PM_{2.5}$ caused by container-cargo trucks and port terminal equipment, Baoshan had the highest population-weighted $PM_{2.5}$ in both January (0.4 μg/m$^3$) and June (0.45 μg/m$^3$) due to its high population and location close to the source (Fig. 8f and 8j). The results of the analyses of different-type shipping-related sources indicated that ship-related sources close to densely-populated areas contribute substantially to population exposures to air pollution

### 3.4 Limitations and uncertainties

Limitations in the study were mainly related to some missing information in database and assumptions during estimation of shipping emissions. When estimating shipping emission inventory, underestimations of actual emissions may be introduced by missing information. For example, AIS data has a high coverage of coastal vessels, but many inland vessels are not equipped with AIS. Emissions from those inland vessels without AIS devices were supplemented by using 2015 vessel call data provided by Shanghai MSA and Shanghai Municipal MSA, and that could introduce some uncertainties for inland river vessels. Also, emissions from fishing boats were probably underestimated because AIS devices on some fishing boats may not be in use. Similarly, limited information exists on auxiliary boilers in the Lloyd's register and CCS databases so we calculated the main engine and auxiliary engine emissions but did not consider auxiliary boiler emissions in this study, which may cause underestimation of shipping emissions.

In addition, we did not consider the external effects of water flow, wind, and waves when calculating engine power for ships going over the region. This would introduce some uncertainties (Aulinger et al., 2016). According to the previous studies in other areas, these factors may increase fuel consumption of individual vessels by as much as 10% to 20%, while the effects of waves on emissions estimations over extensive geographical regions are negligible (Jalkanen et al., 2009; Jalkanen et al., 2012). The downstream of the Yangtze River is located in the geographically

plateau region, and the river flow is below 0.5 m/s (Song and Tian, 1997; Xue et al., 2004). For Shanghai, located at the end of mouth of the Yangtze River to the East China Sea with a flat terrain, the river flow is very slow. Given that ships traveling the Yangtze River near Shanghai have speeds over ground (SOG) of about 5-10 knots (3-5 m/s), the relative ratios of water flow to the SOG is within 20%. In our future work, we will fill the gap in the basic ship data and consider the external effects when building the shipping emission inventory. Finally, this work only extends from emissions to air quality and population exposures. The health impacts of shipping-related air pollution in Shanghai and the YRD region will be explored in future work.

**4 Conclusions**

As the major economic and shipping center in China, the YRD, and in particular Shanghai, experiences high emissions of shipping-related pollutants that result in significant contributions to ambient and population-weighted air pollutant concentrations. Our results showed that on average in 2015 ships contributed 0.75 μg/m$^3$ to the ambient land PM$_{2.5}$ in YRD, with a peak of 4.62 μg/m$^3$ (18.9% of the total ambient PM$_{2.5}$ concentration from all pollution sources) near Shanghai Port. The shipping emissions affecting air quality in the YRD were mainly within 12 NM of shore (over 75% for ship-related SO$_2$ and 50% for ship-related PM$_{2.5}$ concentrations) but emissions coming from 24 to 96 nm offshore also contributed substantially to PM$_{2.5}$ concentrations in the YRD under the transport of summer monsoon. The megacities of Shanghai and Nantong had the highest ship-related population-weighted PM$_{2.5}$ concentrations from the combination of high population density and high shipping emissions. In Shanghai, the inland-water ships contributed a majority (40-80%) of the PM$_{2.5}$ from shipping-related sources; inland-water ships also contributed prominently to population-weighted PM$_{2.5}$ in several districts in Shanghai. These study results on contributions of ships at different distances from shore in the YRD and shipping-related sources in and near Shanghai to ambient air quality and population-weighted PM$_{2.5}$ could inform future ECA policies. For example, policymakers could consider whether to expand China's current DECA boundary of 12 NM to around 100 NM or more to reduce the majority of shipping impacts on air pollution concentrations and exposure. It will be helpful to improve the local air quality and reduce human exposures in densely populated areas by developing more stringent regulations on the fuel quality for ships entering inland rivers or other waterways close to residential regions.

**Author contribution**

YZ and KW conceived the study and made a roadmap for organizing this paper. JF did the air quality simulation and wrote the manuscript. SL ran the shipping emission inventory model. JM ran the WRF model. YZ and CH provided port-related emission inventory. CL and HK provided roadmap for human exposure analysis. AP and WM provided constructive comments in analyzing data. JA and LL provided local-scale land-based emission inventory. YS and JL provided river shipping emission data. XW and QF provided monitoring data. SW and DD provided national land-based emission inventory. JC, WG, and HZ provided container cargo-car traffic emission inventory.

**Acknowledgement**

This research work was partly supported by the National Key Research and Development Program of China (grant no. 2016YFA060130X) and the National Natural Science Foundation of China (21677038). Katherine Walker and Allison Patton give their thanks to the Bloomberg Foundation. The authors wish to thank Noelle Selin and Freda Fung for providing valuable advice. The opinions expressed in this article are the author's own and do not reflect the view of Fudan University, Shanghai Academy of Environmental Science, Shanghai Environmental Monitoring Center, Tsinghua University, Shanghai Urban-rural Construction and Transportation Development Research Institute, or the Health Effects Institute or its sponsors, nor do they necessarily reflect the views and policies of the EPA or motor vehicle and engine manufacturers.

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

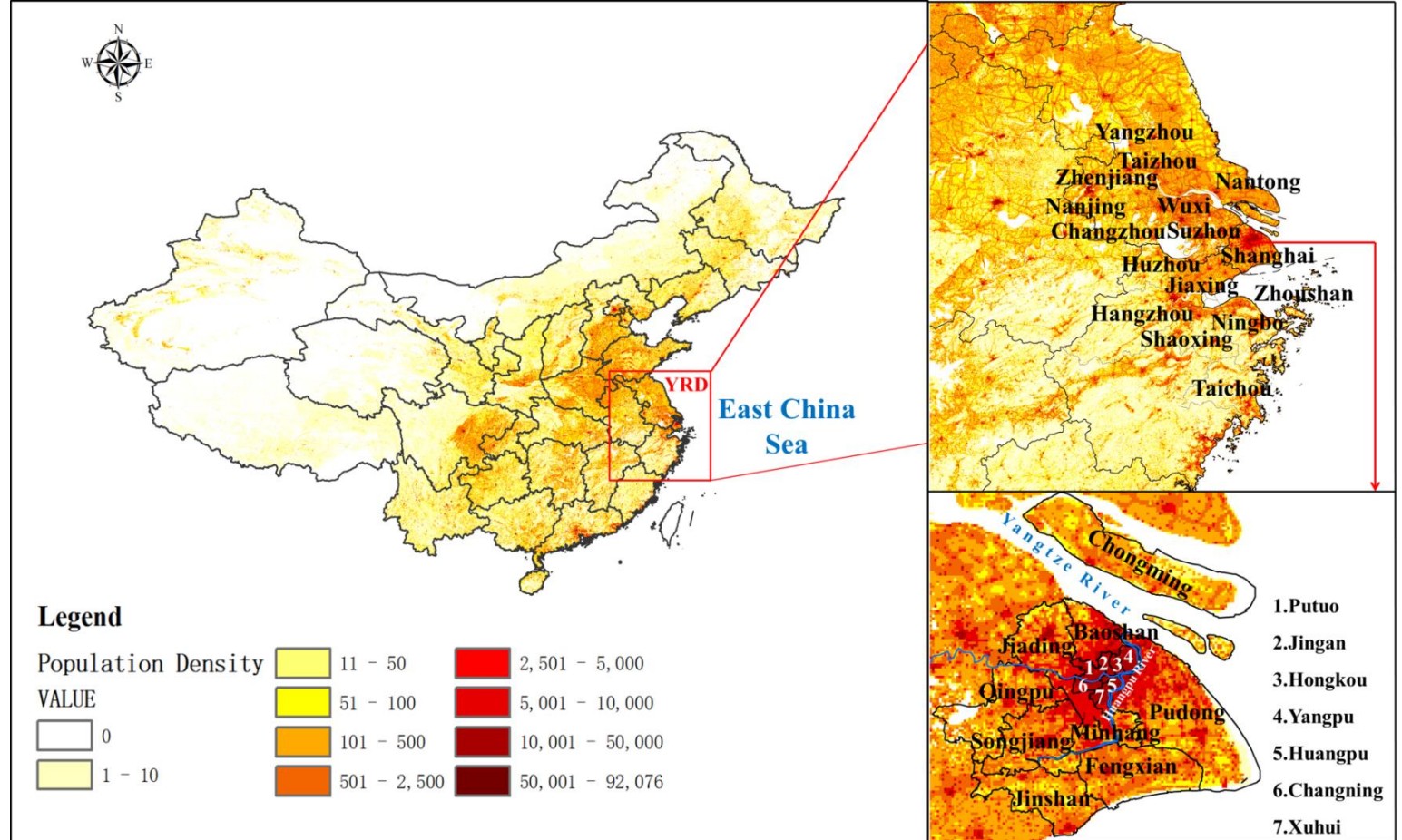

Figure 1. Geographic location of the study area YRD/Shanghai with population density in 2015. 16 core cities in YRD and 16 administrative districts in Shanghai are noted on the map. The smaller administrative districts are labeled with numbers: Putuo (1), Jingan (2), Hongkou (3), Yangpu (4), Huangpu (5), Changning (6), Xuhui (7).

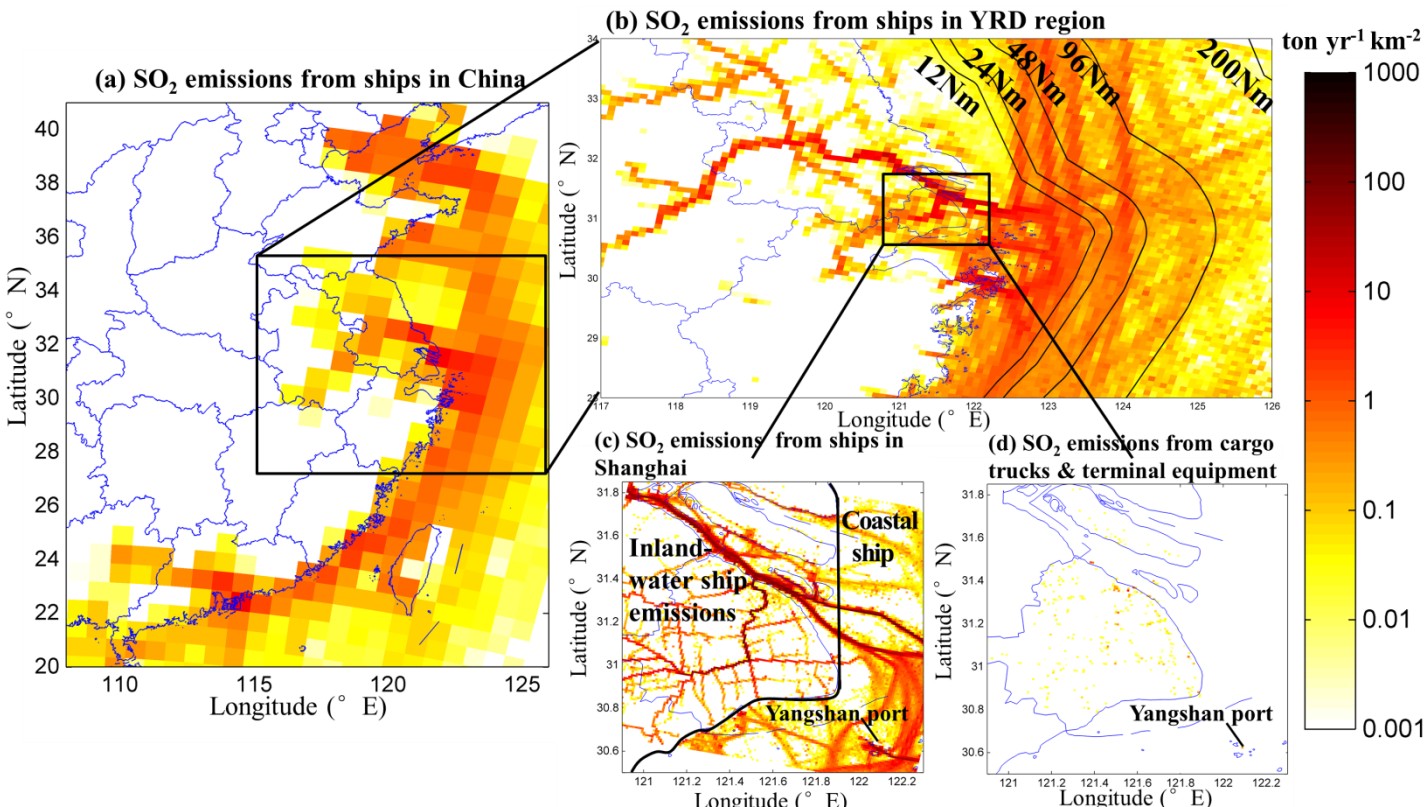

Figure 2. SO$_2$ emissions in 2015 from (a) shipping traffic in China (the average value of January and June) at resolution of 81km $\times$ 81km; (b) ships in different offshore coastal areas (inland-water and within 12 NM, 12-24 NM, 24-48 NM, 48-96 NM and 96-200 NM) in the YRD region, at resolution of 9km $\times$ 9km; (c) inland-water ships and coastal ships in Shanghai, at resolution of 1km $\times$ 1km; and (d) container-cargo trucks and port terminal equipment in Shanghai, at resolution of 1km $\times$ 1km. The black line in (c) refers to the division line between the inland water and coastal water for Megacity Shanghai defined in this study.

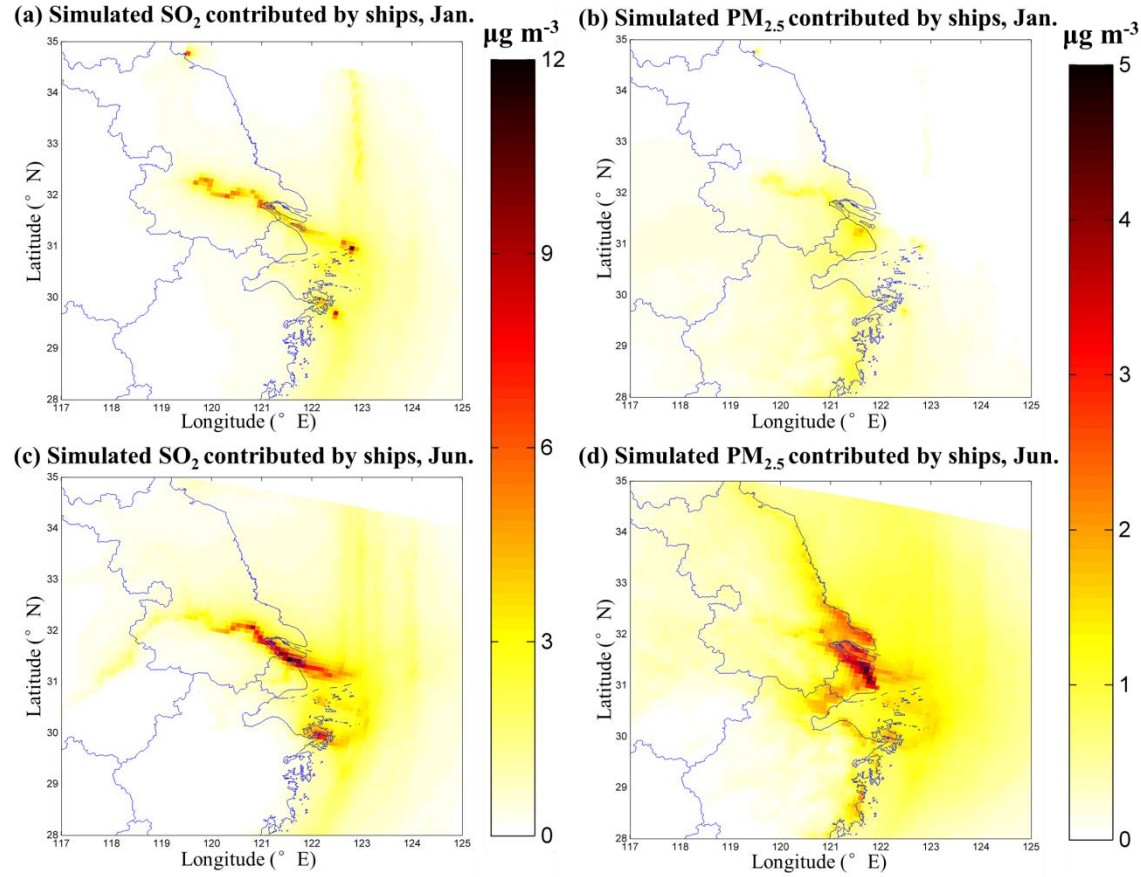

Figure 3. Simulated SO$_2$ (a, c) and PM$_{2.5}$ (b, d) concentrations contributed by shipping

traffic sources in YRD region, in January 2015 (a, b) and June 2015 (c, d)

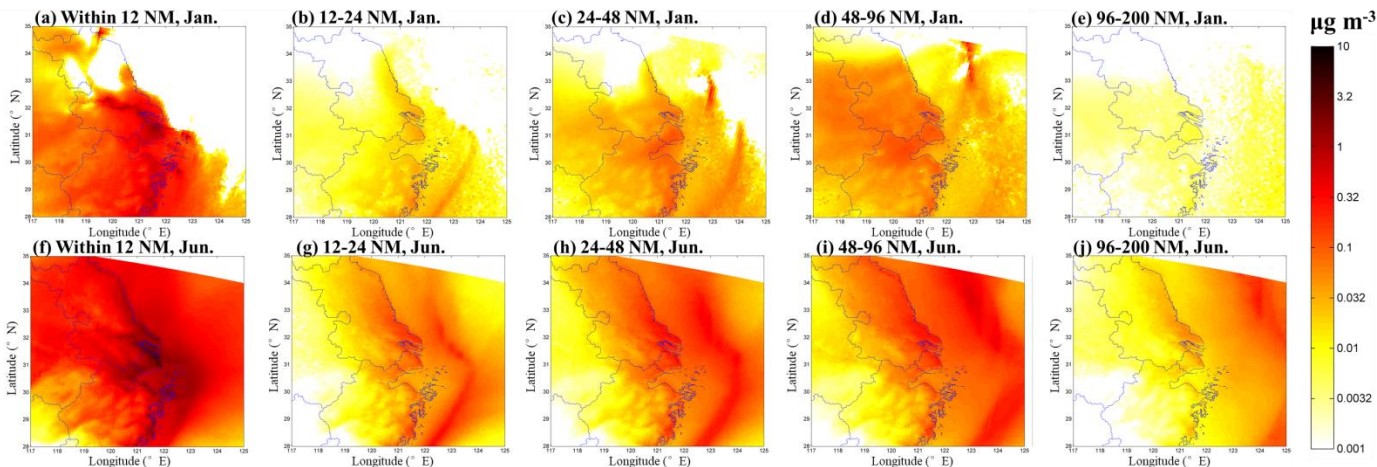

5    Figure 4. Contributions to PM$_{2.5}$ concentrations from shipping emissions at distances

within 12 NM of shore (including inland-waters) (a, f), 12 to 24 NM from shore (b, g),

24 to 48 NM from shore (c, h), 48 to 96 NM from shore (d, i) and 96 to 200 NM from

shore in January 2015 (a-e) and in June 2015 (f-j).

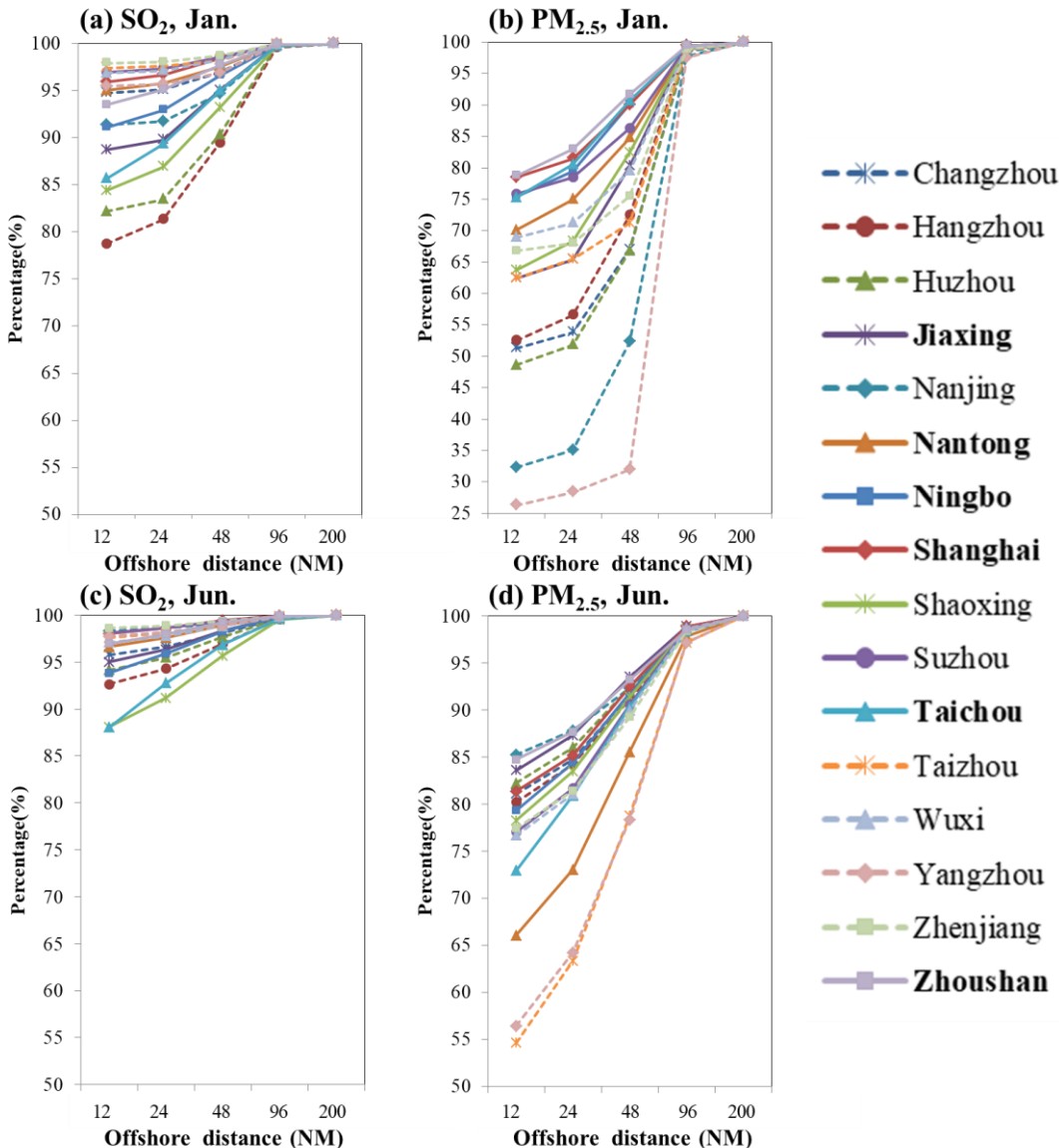

Figure 5. Cumulative contributions of shipping emissions in the YRD at distances within 12 NM of shore (including inland-waters), 24 NM from shore, 48 NM from shore, 96 NM from shore, and 200 NM from shore to $PM_{2.5}$ concentrations (a, c) and $SO_2$ concentrations (b, d) in January 2015 (a, b) and in June 2015 (c, d). Names of Coastal cities are bold in the legend.

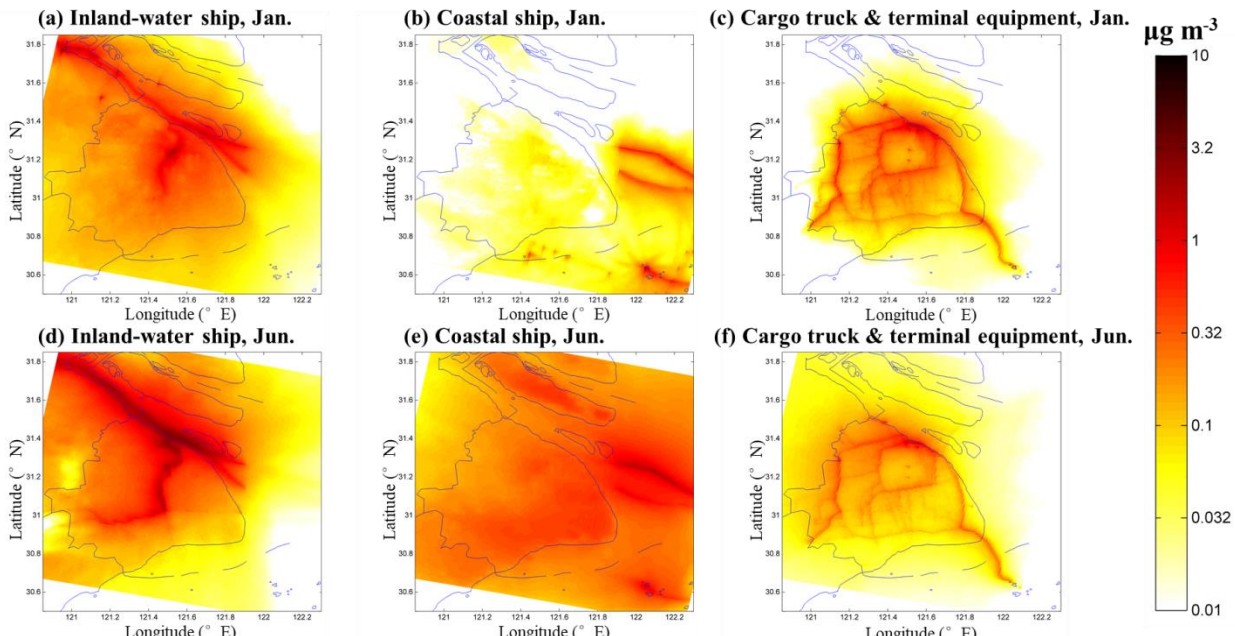

Figure 6. Contributions to PM$_{2.5}$ concentrations from inland-water ships (a, d), coastal ships (b, e) and container-cargo trucks and port terminal equipment (c, f) in January 2015 (a-c) and June 2015 (d-f).

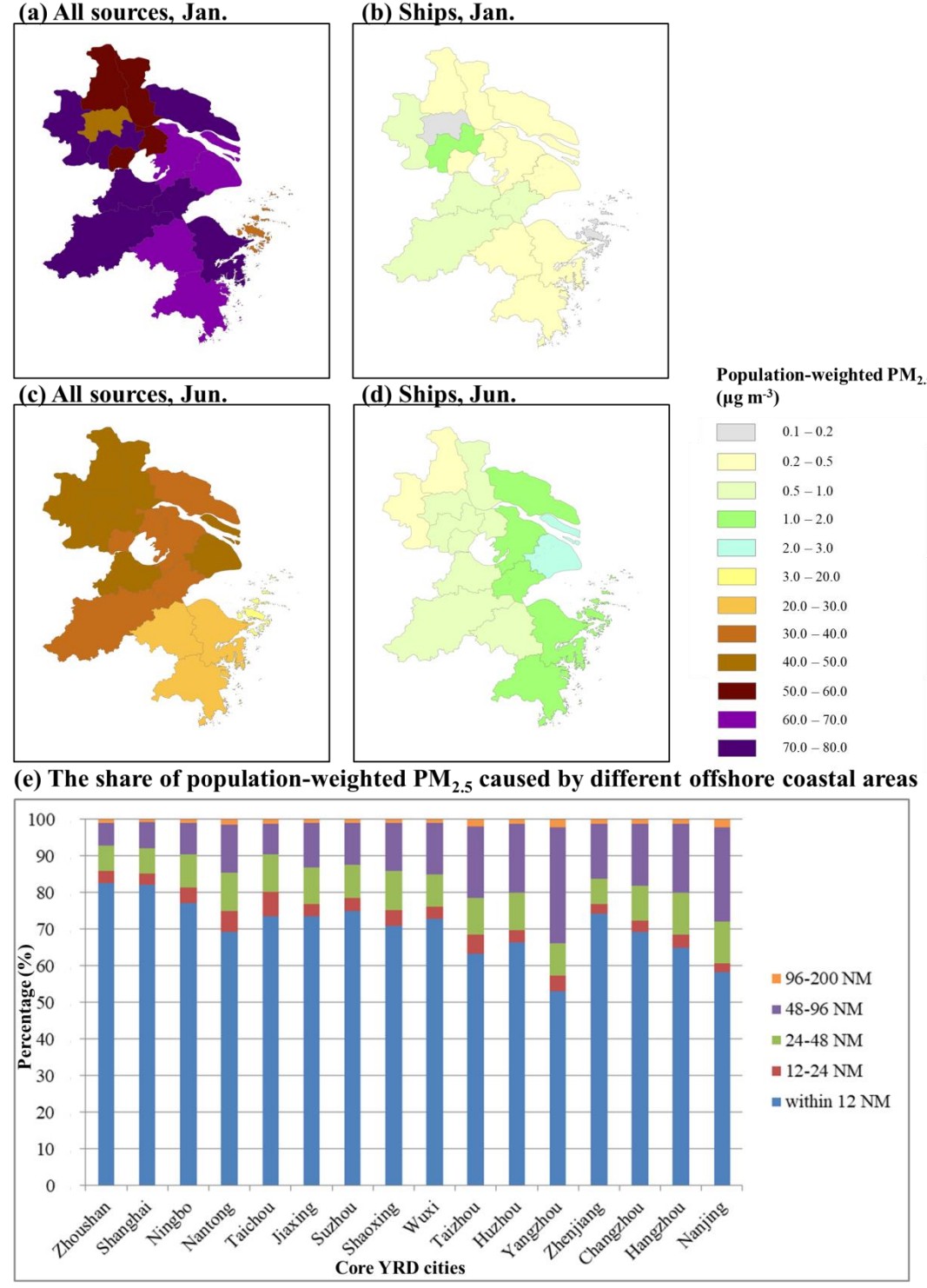

Figure 7. The spatial distribution of population-weighted PM$_{2.5}$ in 16 YRD cities caused by all pollution sources (a, c) and by all ships (b, d) in January 2015 (a, b) and June 2015 (c, d); the average share of population-weighted PM$_{2.5}$ in 16 YRD cities caused by different offshore coastal areas in all ships (e). The cities' names are ordered by their distance to the coast.

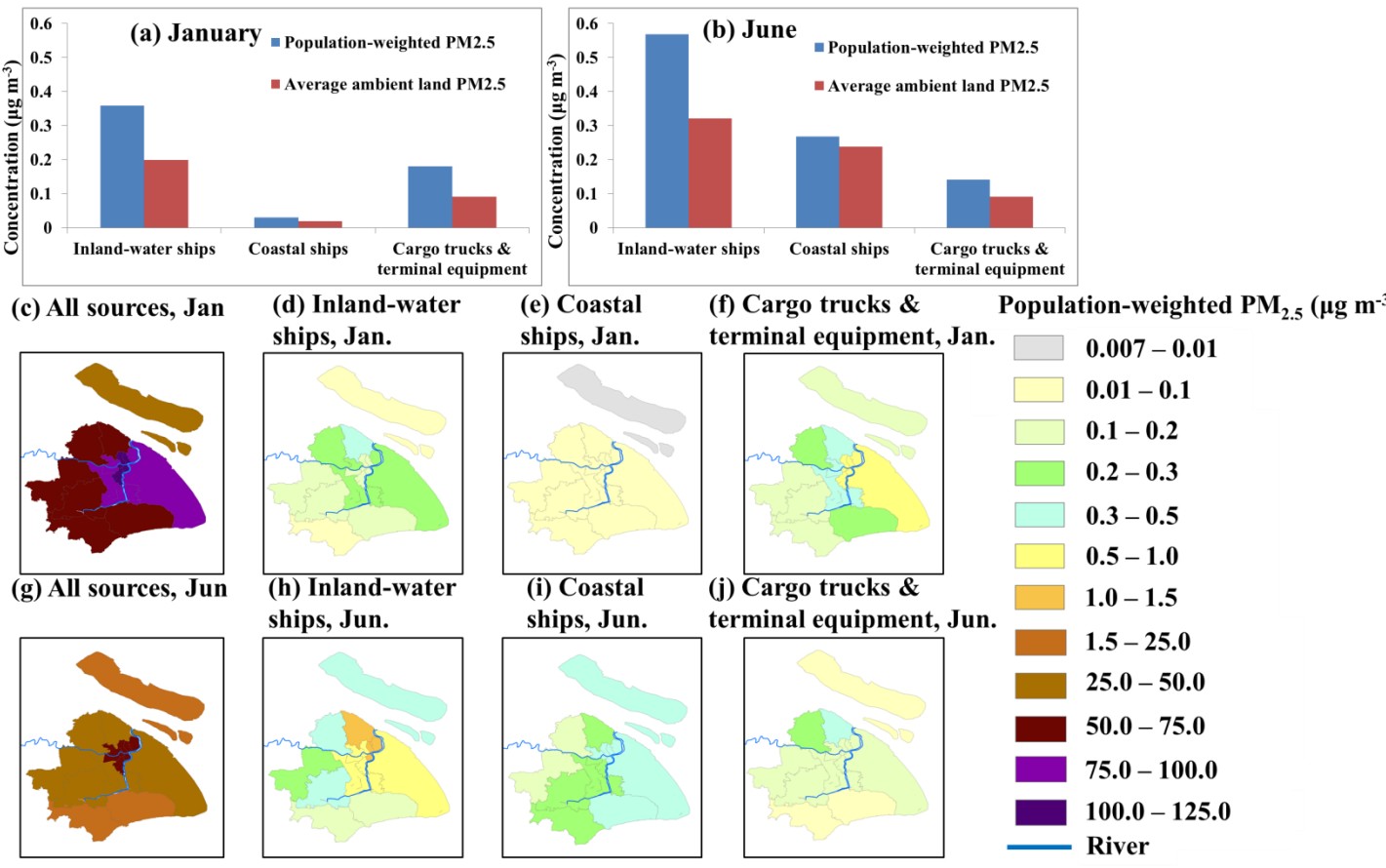

Figure 8. Population-weighted PM$_{2.5}$ and average PM$_{2.5}$ caused by different ship-related sources in Shanghai, in January(a) and in June (b); population-weighted PM$_{2.5}$ caused by all pollution sources (c, g), inland-water ships (d, h), coastal ships (e, i) and container-cargo trucks and port terminal equipment (f, j) in 16 districts in Shanghai, in January 2015(c-f) and June 2015 (g-j).

Table 1 Statistical metrics of the model evaluation. Observed data (Obs.) and simulated data (Sim.) for each city are the average of monthly values of January and June case. NMB, NME, RMSE and $r$ were calculated based on the daily-average observed and simulated data.

| City | SO$_2$ | | | | | | PM$_{2.5}$ | | | | | |
| | Obs. | Sim. | NMB (%) | NME (%) | RMSE ($\mu g\ m^{-3}$) | $r$ | Obs. | Sim. | NMB (%) | NME (%) | RMSE ($\mu g\ m^{-3}$) | $r$ |
|---|---|---|---|---|---|---|---|---|---|---|---|---|
| Changzhou | 31.24 | 20.14 | -35.55 | 40.85 | 15.79 | 0.80 | 74.21 | 68.27 | -8.01 | 32.51 | 31.99 | 0.76 |
| Hangzhou | 16.84 | 13.75 | -18.35 | 28.74 | 6.77 | 0.83 | 59.35 | 56.96 | -4.03 | 28.21 | 22.05 | 0.75 |
| Huzhou | 19.25 | 14.73 | -23.52 | 38.81 | 11.45 | 0.80 | 65.13 | 70.50 | 8.25 | 45.60 | 39.17 | 0.47 |
| Jiaxing | 25.37 | 16.84 | -33.67 | 50.58 | 17.31 | 0.75 | 61.31 | 57.01 | -7.02 | 33.98 | 29.96 | 0.65 |
| Nanjing | 22.39 | 16.38 | -20.60 | 26.50 | 10.13 | 0.76 | 68.20 | 55.71 | -14.06 | 27.80 | 32.30 | 0.60 |
| Nantong | 32.73 | 22.05 | -32.66 | 49.69 | 23.21 | 0.70 | 68.69 | 51.15 | -25.54 | 39.27 | 37.23 | 0.69 |
| Ningbo | 16.20 | 10.47 | -35.42 | 42.01 | 7.64 | 0.83 | 55.47 | 48.06 | -13.37 | 34.51 | 28.49 | 0.75 |
| Shanghai | 19.16 | 12.32 | -35.72 | 40.23 | 10.72 | 0.83 | 63.64 | 67.77 | 6.50 | 36.18 | 28.71 | 0.75 |
| Shaoxing | 22.47 | 14.63 | -34.91 | 40.03 | 10.36 | 0.80 | 61.90 | 56.86 | -8.15 | 34.06 | 27.21 | 0.70 |
| Suzhou | 21.37 | 15.16 | -29.09 | 37.26 | 10.39 | 0.85 | 67.11 | 56.45 | -15.89 | 33.41 | 28.76 | 0.76 |
| Taichou | 10.72 | 7.55 | -29.64 | 34.07 | 5.25 | 0.80 | 47.55 | 43.69 | -8.11 | 35.35 | 24.09 | 0.52 |
| Taizhou | 29.64 | 20.84 | -29.70 | 61.53 | 22.63 | 0.67 | 74.56 | 62.82 | -15.75 | 31.75 | 33.49 | 0.63 |
| Wuxi | 24.64 | 18.89 | -23.35 | 30.85 | 10.58 | 0.87 | 73.45 | 59.36 | -19.20 | 31.80 | 30.92 | 0.77 |
| Yangzhou | 25.78 | 18.75 | -27.31 | 44.22 | 15.17 | 0.62 | 62.30 | 60.12 | -3.50 | 46.10 | 37.08 | 0.57 |
| Zhenjiang | 29.65 | 21.50 | -27.51 | 39.49 | 16.23 | 0.61 | 67.78 | 62.61 | -7.63 | 33.88 | 30.31 | 0.59 |
| Zhoushan | 9.99 | 8.04 | -19.60 | 40.42 | 6.73 | 0.64 | 30.13 | 19.81 | -34.28 | 49.15 | 16.82 | 0.78 |

Table 2. Primary emissions (ton/yr), emission share in all shipping emissions (%) and emissions density (ton/yr/km$^2$) from shipping at different boundaries in YRD region[a] in 2015

| | Pollutants | Within 12 NM | 12-24 NM | 24-48 NM | 48-96 NM | 96-200 NM |
|---|---|---|---|---|---|---|
| Shipping emission inventory (ton/yr) | SO$_2$ | $1.3\times10^5$ | $1.4\times10^4$ | $2.5\times10^4$ | $3.2\times10^4$ | $1.3\times10^4$ |
| | NO$_x$ | $3.6\times10^5$ | $2.0\times10^4$ | $3.5\times10^4$ | $4.5\times10^4$ | $1.8\times10^4$ |
| | PM$_{2.5}$ | $1.3\times10^4$ | $2.4\times10^3$ | $4.5\times10^3$ | $5.4\times10^3$ | $1.5\times10^3$ |
| | VOC$_s$ | $7.9\times10^3$ | $8.3\times10^2$ | $1.3\times10^3$ | $1.5\times10^3$ | $3.0\times10^2$ |
| Emission share in all shipping emission (%) | SO$_2$ | 61.4 | 6.4 | 11.4 | 14.9 | 5.8 |
| | NO$_x$ | 75.0 | 4.1 | 7.4 | 9.6 | 3.9 |
| | PM$_{2.5}$ | 48.4 | 9.0 | 16.9 | 20.2 | 5.5 |
| | VOC$_s$ | 66.6 | 7.0 | 11.2 | 12.6 | 2.6 |
| Emission density (ton/yr/km$^2$) | SO$_2$ | 0.66 | 0.54 | 0.49 | 0.33 | 0.06 |
| | NO$_x$ | 1.74 | 0.86 | 0.77 | 0.51 | 0.08 |
| | PM$_{2.5}$ | 0.08 | 0.06 | 0.06 | 0.04 | 0.01 |
| | VOC | 0.05 | 0.02 | 0.01 | 0.01 | 0.001 |

5  a.  domain 3

Table 3. Primary emissions (ton/yr) and emission share in all pollution sources (%) from different-type ship-related source in Shanghai[a] in 2015

| | Ship-related source | SO$_2$ | NO$_x$ | PM$_{2.5}$ | VOC |
|---|---|---|---|---|---|
| Emission inventory (ton/yr) | Inland-water ships[b] | $3.3\times10^4$ | $9.2\times10^4$ | $0.40\times10^4$ | $0.27\times10^4$ |
| | Coastal ships[c] | $1.6\times10^4$ | $2.9\times10^4$ | $0.18\times10^4$ | $0.067\times10^4$ |
| | Container-cargo trucks | 0.0 | $1.8\times10^4$ | $0.064\times10^4$ | $0.11\times10^4$ |
| | Port terminal equipment[d] | $0.0021\times10^4$ | $0.18\times10^4$ | $0.0057\times10^4$ | $0.022\times10^4$ |
| Emission share in all pollution sources in Shanghai (%) | Inland-water ships | 11.8 | 18.7 | 3.6 | 0.5 |
| | Coastal ships | 5.6 | 5.8 | 1.6 | 0.1 |
| | Container-cargo trucks | 0.0 | 3.7 | 0.6 | 0.2 |
| | Port terminal equipment | 0.01 | 0.36 | 0.05 | 0.04 |

a. domain 4

10  b. defined as ships operate in both the outer port and in the inner river region of Shanghai Port, which include Yangtze River, Huangpu River and other river ways in Shanghai

c. includes China coastal and international ships

d. includes cranes and forklifts used for internal transport