# Peer review of "The influence of spatiality on shipping emissions, air quality and potential human exposure in Yangtze River Delta/Shanghai, China"

_Atmospheric Chemistry and Physics, 2018_

## Referee Comment (RC1) · Anonymous Referee #1 · 14 Jan 2019

General comments:

The manuscript of Feng et al, "The influence of spatiality on shipping emissions, air quality and potential human exposure in Yangtze River Delta/Shanghai, China", is well written and provides some additional information on the spatial distribution of ship emissions of the inland waterway traffic. This manuscript feels like an attempt to achieve something greater in the future, because it introduces the methodology necessary for ship emission inventory work, atmospheric modeling and health effect evaluation without getting there in the end. The title wisely stops at human exposure, because this is what the paper delivers, but I wonder why the authors stopped there and did not take the final step from exposure to health effects.

The novelty aspect of this work could be improved; emission inventory work cites existing work and this paper does not bring much new to this topic. The atmospheric modeling was done with an existing code and no advances were made to improve the existing tools. From methodological point of view, this paper applies existing tools to a known environmental problem which means that the novelty must come from that contribution. There are two contributions which are brought to light in this paper. First is the contribution of inland waterway traffic to ship emissions and the second is the geographical reach of ship emissions when ship to shore distance is varied. The latter contribution hints to a design of new potential regulation which would not necessarily cover all of the 200 nautical mile distance from shore, but this motivation is currently only indirectly stated, if at all.

In some parts of the manuscript, authors state that they have used data from specific months whereas on other parts data for a full year seems to be used. It was challenging to understand which parts of the work were done with a full year's dataset and which with less data.

Detailed comments:

Page 1, Introduction, lines11-15. Authors discuss the health effect evaluation of ship emissions and quote Sofiev et al (2018). I wonder, what is the motivation of not citing the numbers of Sofiev et al, which reports the latest global health effect numbers, but authors choose to refer to 50 000 to 90 000 premature mortality cases instead? The values given in Sofiev et al (2018) are much higher than this.

Page 4, lines 24-29. Authors have chosen to report the case before the DECA implementation. I was wondering about the motivation of this decision, because it seems that the modeling work could have been easily applied also the DECA case and would have allowed the identification of the impacts of this policy change thus significantly improving the novelty aspect of this work.

Page 5, lines 15-22: Authors report the specifics of chemical transport model domains, but say very little of the emissions. There is a separate section for ship emissions, but I cannot see whether daily, monthly or annual emissions with or without the dynamic features of ship emissions were used or not. The activity data allows this, but have the authors considered these variations in to consecutive steps, too?

Page 5, lines 23-27. "Highest shipping impacts were expected in June because shipping activity and emissions are higher in summer than at other times of the year". There are references to Fan et al

(2016) and Jalkanen (2009) in this sentence. Actually Fan et al state "No significant differences in the total emissions quantities were observed among summer, autumn and winter", which seems to contradict what the authors say.

Page 6, lines 1-2. I would like to see some discussion on the limitations of AIS in this. It is not used by all ships listed by the authors. The way the text is written now implies that all the ship classes listed here is covered by AIS, which is not necessarily the case since inland traffic may be incompletely represented.

Page 6, lines 4-6. Does the material obtained from MSA include boats? I would imagine that boats outnumber ships by at least an order of magnitude. Was any consideration given to boat contributions to emissions? Boats may not be the biggest source of CO2, NOx or SOx, but they are a significant source of VOCs and CO.

Page 6, lines 7-12. Use of speed entries of AIS. How did you count for the water flow? You have concentrated the study on an area which is along a large river, which means that there is a significant water flow. When a case like this occurs, speed over water is not the same as the speed over ground indicated by the AIS. If power predictions are based on speed over ground, then power prediction will fail. Have the authors considered this aspect?

Page 6, lines 13-14. This is a rather drastic assumption. Have you thought about linking the fuel type or S content to engine specifications? There are technical reasons why some engines cannot use certain types of fuels, but have authors chosen to neglect these limitations completely?

Page 7, lines 3-6. "Shopping" of emission data piece by piece from various data providers may lead to unexpected side effects, which can arise from the fundamental assumptions used in emission inventory construction work. The CO and VOC emissions both result from incomplete combustion of fuel and there is a high probability that these two are linked. Did the authors check what the CO/VOC share in IIASA inventory was and how different the CO/VOC share was in the combined inventory?

Page 7, lines 16-21. What was the temporal resolution of the ship emission inventories used in this work?

Page 8, lines 15-17, the last sentence. There is no uncertainty involved in atmospheric measurements? Really? These can be tens of percent, easily. Cross comparisons of AQ measurement results between instruments can deviate significantly, depending on the equipment used.

Page 8, lines 25-29. I agree that population weighted approach has some merit, but that still is an incomplete representation of human activity. The approach used here assumes that people spend all their time at home and do not consider realistic behavior of people. There are some studies that take this into account (see for example Soares et al, GMD, 2014).

Page 9, lines 1-2. It seems that the annual estimate is based on two months of actual data. Why not using data for the whole year? This would remove one source of uncertainty from the final results. The lines 9-10 seem to suggest that data for the whole year 2015 was available for the authors.

Page 9, lines 18-20. The largest contribution to emissions comes from sources close to the shore. This underlines the importance of including all waterborne traffic sources and consideration of water flow/speed issue. Some discussion of these topics should be included in the manuscript.

Page 10, lines 8-10. Authors identify ships as a significant source of VOCs. Have you considered the role of small boats in VOC emissions? The VOC emission levels allowed for boat engines are significantly higher than those of marine diesel engines and there are a lot of small engines in boats.

Page 12, lines 19-21. If the atmospheric conversion of gaseous $SO_2$ to particulate $SO_4$ takes about a week (reacts with OH), why is the 12 nm distance relevant in this aspect? Surely during one week the gaseous $SO_2$ travels further that 12 nm during that time and it cannot be used as an only explanation why ships further out than 12 nm do not contribute to $SO_2$.

Figure 1
The legend text font size should be increased, it is very small reading as it is now. It is especially tough to read the text of the right hand side zoomed images.

Figure 2. This figure is confusing. If the symbols represent measurement values, I cannot see any numerical values linked to the symbols. If the colors correspond to gridded model concentrations, that is fine, but the measured values cannot be determined from these images. Perhaps another form of graphic could be used to provide the comparisons?

Figure 3. The legend texts are very small in this figure, too.

Figure 5, I would welcome some discussion why the distance to the shore is relevant in this context. Are the authors trying to see whether it is useful to limit the distance of regulated emissions to a specific value or what is the reasoning of choosing these distance bins?

Figure 9. Texts are too small, especially in the two top images.

Table 1. Are these daily, monthly or annual values? There is no indication of the timeline here? This data does not tell me very much of how well the model is able to capture the temporal variability of pollution peaks. Could a line graph be used here instead? This would help to see how well the model is able to capture the air concentrations.

Table 2. No units are given?

Table 3. No units given?

Supplementary material, S1

The authors seem to apply the Starcrest methodology in their emission modeling.

Page 2, text under Eq (1). Maximum speed and design speed of ships are two different things and IHS data often mentions economic speed. Which was one was actually used in the analysis?

Page 2, near Eq (4), aux boiler use. Did the authors consider the exhaust boilers at all in this regard? Also, the installed boiler capacity is difficult to determine from ship databases, because this field is not provided. I would like to know where the installed boiler data comes from.

Page 2, last paragraph. Authors make reference to Lloyds, 2009 which is not listed in the bibliography provided for S1. Also, why refer to data from 2009 if the AIS data is for 2015. How were the ships built during 2009-2015 treated?

Page 2, last paragraph. Authors assume all inland waterway vessels to have 7000 kW engine? No effort was made to identify these vessels and use proper description of installed power?

Page 3, S2, first paragraph. "Table S1 lists emission factors used in the present study". This is not true and the emission factor table is missing.

Page 3, second paragraph. Add reference ICF, 2009

Page 3, second paragraph. OC and EC low load adjustment factors were treated the same way as PM. This is contrary to the behavior of EC and EC as a function of engine load. Authors might want to check the ICCT report "Black Carbon Emissions and Fuel Use in Global Shipping, 2015", Oct 2017 for low load behavior of carbon fraction.

---

## Referee Comment (RC2) · Anonymous Referee #2 · 28 Jan 2019

General comments: This study presented the importance of geographical locations of ship emissions to the environmental and human health effects. The manuscript has been well written and organized. Take the YRD region– one of the busiest port cluster in the world as the example, this study result is helpful to understand the meaningful points of future ECA policy. The authors should explicit the key implication through the paper, including the abstract, result and conclusion part. Also, there are some minor details should be improved.

The details should be improved: Page6~7, 2.2.2 Non-shipping emission inventories part For the national scale domain and regional scale domain, several sets emission

data has been used. The authors should make clearer how they merge the emission together. How did they use 2015 national emission database to make a regional 27 km × 27 km resolution that included 5 pollutants? Did they use spatial interpolation method? Which year are the IIASA data for CO and NH3? Page7ïijŽLine 15∼16ïijŽ "The initial and boundary conditions for meteorology were generated from the Chinese National Centers for Environmental Prediction (NCEP) Final Analysis (FNL) "ïijŇhere the authors should confirm the NCEP FNL data source.

Page 9, line 12-17: The authors compared the result of YRD shipping emission with Fan et al.'s and Chen et al.'s studies. The authors quoted Liu et al. (2018) to compare the proportion of YRD shipping emissions in whole China. However, Liu et al. (2018) also reported YRD shipping emissions. Why not compare the result with the values in Liu et al. (2018) as well?

Page 10, line 12-16: The authors quoted Fu et al. (2012), which used 2010 vessel call data to estimate shipping emissions. I suggest authors reviewed recent studies using AIS data to make comparisons in Shanghai port.

Page 12, line 6-15: The contribution to SO2 from ships in different coastal areas was not discussed in this paragraph. But in the following paragraph, the authors discussed cumulative contributions from ships at different distance to both SO2 and PM2.5. It shows no consistency when authors discussed SO2 results throughout the section 3.2.2.

Page 14, line 1-6: The authors discussed the population-weighted PM2.5 from both shipping source and all pollution sources. Then, what's the proportion of population-weighted PM2.5 from the shipping source among all pollution sources? I suggest some discussion here.

Page 15, line 25: The uncertainty analysis is lacked in the section of result and discussion. The uncertainties of shipping emission inventories should be discussed here.

---

## Author Comment (AC1) · 9 Mar 2019

Dear Editor and Referees,

We are pleased to submit our responses to all the comments and revision for manuscript acp-2018-1163. We appreciate all the comments and suggestions that are especially helpful. All the referees' comments have been addressed carefully.

Best regards with respect,
Yan Zhang, representing all co-authors

Reviewers' comments are in blue.
Authors' responses are in black.
Revisions in manuscript are in *italic*, underlined.

General comments:
1. The manuscript of Feng et al, "The influence of spatiality on shipping emissions, air quality and potential human exposure in Yangtze River Delta/Shanghai, China", is well written and provides some additional information on the spatial distribution of ship emissions of the inland waterway traffic. This manuscript feels like an attempt to achieve something greater in the future, because it introduces the methodology necessary for ship emission inventory work, atmospheric modeling and health effect evaluation without getting there in the end. The title wisely stops at human exposure, because this is what the paper delivers, but I wonder why the authors stopped there and did not take the final step from exposure to health effects.
**Response:**
We thank the reviewer for this comment. As the reviewer has noted, the focus of this paper was on the results of our approach to estimating the impact of shipping and related activities on $PM_{2.5}$ concentrations and where those concentrations differed when examined in light of where the population lives. We do intend to prepare a manuscript that examines the health impacts of these exposures in detail, and have added to the text that "this work only extends from emissions to air quality and population exposures. The health impacts of shipping-related air pollution in Shanghai and the YRD region will be explored in future work."
**Revisions in the manuscript:**
*1. Page 20, line 7-9: "Finally, this work only extends from emissions to air quality and population exposures. The health impacts of shipping-related air pollution in Shanghai and the YRD region will be explored in future work."*

2. The novelty aspect of this work could be improved; emission inventory work cites existing work and this paper does not bring much new to this topic. The atmospheric modeling was done with an existing code and no advances were made to improve the existing tools. From methodological point of view, this paper applies existing tools to a known environmental problem which means that the novelty must come from that

contribution. There are two contributions which are brought to light in this paper. First is the contribution of inland waterway traffic to ship emissions and the second is the geographical reach of ship emissions when ship to shore distance is varied. The latter contribution hints to a design of new potential regulation which would not necessarily cover all of the 200 nautical mile distance from shore, but this motivation is currently only indirectly stated, if at all.

**Response:**

Thank you for the suggestions to more clearly identify the novelty of our work, especially regarding the contribution of inland waterway traffic to ship emissions and the spatial distribution of ship emissions, ambient pollutant concentrations, and human exposures. We have expanded on the description of the gap in the literature that this paper addresses and the policy implications of the research results throughout the manuscript. Also, it is also novel that both of the regional and port scale influences of shipping emission on air quality have been considered in this manuscript.

**Revisions in the manuscript:**

*1. Page 2, line 6-8:* "in particular, in the YRD region, expanding the boundary of 12 NM in China's current DECA policy to around 100 NM would include most of the shipping emissions affecting air pollutant exposures, and stricter fuel standards could be considered for the ships on inland rivers and other waterways close to residential regions."

*2. Page 5, lines 6-15:* "In China, a few studies reported the contribution to air pollution from shipping in different offshore coastal areas or individual ship-related sources. For example, Mao et al. (2017) estimated primary emissions from OGVs at different boundaries in the PRD region, and concluded that further expansion of emission control area to 100 NM would provide even greater benefits. However, the impacts of shipping emissions at varying distances from shore on air quality and potential human exposure, which are important when considering ECA policy, have not been rigorously studied. Mao and Rutherford (2018) studied $NO_x$ emissions from three categories of merchant vessels—OGVs, coastal vessels (CVs) and river vessels (RVs) in China's coastal region. But less attention was paid to the impacts of inland waterway traffic and port-related sources like container-cargo trucks and terminal port equipment on air quality and potential human exposure."

*3. Page 5, line 30; page 6, line 1-3:* "The results of this study could be informative to the consideration of the distance of regulated emissions in the design of future emissions control areas for shipping in YRD, or regulations on the sulfur content of fuels for individual ship-related sources in Shanghai."

*4. Page 16, line 7-10:* "The results of these YRD analyses suggest that although ambient ship-related $SO_2$ concentrations were mainly affected by shipping inland or within 12 NM, expanding China's current DECA to around 100 NM or more would reduce the majority of the impacts of shipping on regional $PM_{2.5}$ pollution."

*5. Page 19, line 10-12:* "The results of the analyses of individual shipping-related sources indicated that ship-related sources close to densely-populated areas contribute substantially to population exposures to air pollution."

*6. Page 20, line 26-29; page 21, line 1-2:* "For example, policymakers could consider

*whether to expand China's current DECA boundary of 12 NM to around 100 NM or more to reduce the majority of shipping impacts on air pollution and exposure or to develop more stringent regulations on the sulfur content of fuels for ships entering inland rivers or other waterways close to residential regions due to their significant influence on local air quality and human exposures in densely populated areas."*

3. In some parts of the manuscript, authors state that they have used data from specific months whereas on other parts data for a full year seems to be used. It was challenging to understand which parts of the work were done with a full year's dataset and which with less data.

**Response:**

Thank you for pointing out this problem. We have revised the relevant parts of the manuscript to clarify the temporal scope of data used in our work. It is due to the limitation of getting the national-scale AIS data for the whole year from the marine-time department, only data in some representative month like January and June are available for our study. Therefore, we used the average values of these two months to estimate annual shipping emissions in whole China. But we have full-year AIS data in Yangtze River Delta (YRD), and the estimates of annual shipping emissions in YRD scale and Shanghai city scale in the manuscript were based on the full-year data. To identify the impact of shipping on ambient air quality and population exposure, January and June were selected as representative months to conduct sensitivity experiments, and monthly shipping emissions for January and June were used in the air quality model. We have clarified the time scale of data we used in the revised manuscript.

**Revisions in the manuscript:**

*1. Page 5, line 18-28: "We modeled shipping emissions in different offshore areas in the YRD region and emissions from individual ship-related sources in Shanghai city for each month of the year. To identify which offshore areas in the YRD region and which individual ship-related sources in Shanghai contributed the most ambient air pollution, and human population exposure, we modeled the impacts of shipping emissions in different offshore areas (within 12 NM including inland waters, 12-24 NM, 24-48 NM, 48-96 NM, and 96-200 NM) in the YRD region as well as coastal ships, inland-water ships, and container-cargo trucks and port terminal equipment in and near the port areas under the jurisdiction of Shanghai MSA in two representative months (January and June)."*

*1. Page 7, line 12-14: "Emissions from ships entering the geographic domains for YRD or Shanghai were calculated using the AIS-based model developed by Fan et al. (Fan et al., 2016), and monthly shipping emissions for January and June were used in the air quality model to capture the seasonal variation to expect more accurately than annual shipping emissions with no monthly variations."*

*2. Page 11, line 12-14: "Due to limitation of the data source, the national-scale AIS data in this study only covered the representative months of January and June 2015, while the YRD-scale AIS data covered 2015 full year."*

Detailed comments:

4. Page 1, Introduction, lines11-15. Authors discuss the health effect evaluation of ship emissions and quote Sofiev et al (2018). I wonder, what is the motivation of not citing the numbers of Sofiev et al, which reports the latest global health effect numbers, but authors choose to refer to 50 000 to 90 000 premature mortality cases instead? The values given in Sofiev et al (2018) are much higher than this.

**Response:**

Thank you for the question. Previously, we more focused on the impact of shipping in past years, so that the numbers we referred to are the estimates of past years (2010 and 2012). Based on your reminding, we think it is better to also cite the value in Sofiev et al (2018), which reports a 2020 projection of shipping's impact, so as to give a more comprehensive review of the health effect evaluation of ship emissions.

**Revisions in the manuscript:**

*1. Page 2, line 19-24: "Globally, about 50,000 to 90,000 cardiopulmonary diseases and lung cancer deaths were attributable to exposure to particulate matter emitted from shipping in 2010 and 2012, respectively (Corbett et al., 2007;Partanen et al., 2013;Winebrake et al., 2009), and 403,300 premature mortalities per year due to shipping are predicted in 2020 under business-as-usual (BAU) assumptions (Sofiev et al., 2018)."*

5. Page 4, lines 24-29. Authors have chosen to report the case before the DECA implementation. I was wondering about the motivation of this decision, because it seems that the modeling work could have been easily applied also the DECA case and would have allowed the identification of the impacts of this policy change thus significantly improving the novelty aspect of this work.

**Response:**

Thank you for the question. This study aimed to evaluate the impact of shipping emissions on air quality prior to implementation of the DECA policy as the baseline. Taking 2015 as the baseline year can reflect the situation for recent years. Also, this research aimed to provide basic scientific evidence to inform policies for controlling future shipping emissions. The first-phase DECA policy during 2016-2018 only applied to ships during berthing at port. Evaluation of potential future DECA policies will be done in ongoing work.

6. Page 5, lines 15-22: Authors report the specifics of chemical transport model domains, but say very little of the emissions. There is a separate section for ship emissions, but I cannot see whether daily, monthly or annual emissions with or without the dynamic features of ship emissions were used or not. The activity data allows this, but have the authors considered these variations in to consecutive steps, too?

**Response:**

Thank you for the question. The ship emissions were actually calculated at 5-minute intervals based on AIS data. Then we used the monthly dynamic ship emissions as the input to the air quality modelling since our air quality analysis has been based on the

monthly time scale. Also the monthly mean simulation results were evaluated in this study to show they can match with the observations well. We've clarified the use of monthly shipping emission in the air quality model in the revised manuscript.

**Revisions in the manuscript:**
*1. Page 7, line 12-14: "Emissions from ships entering the geographic domains for YRD or Shanghai were calculated using the AIS-based model developed by Fan et al. (Fan et al., 2016), and monthly shipping emissions for January and June were used in the air quality model to capture the seasonal variation to expect more accurately than annual shipping emissions with no monthly variations."*

7. Page 5, lines 23-27. "Highest shipping impacts were expected in June because shipping activity and emissions are higher in summer than at other times of the year". There are references to Fan et al (2016) and Jalkanen (2009) in this sentence. Actually Fan et al state "No significant differences in the total emissions quantities were observed among summer, autumn and winter", which seems to contradict what the authors say.

**Response:**
Thank you for the question. In this study, shipping emissions in summer were slightly higher than the other seasons (The ship emission in summer accounted for more than 28% of the annual shipping emissions, a little higher than other season). In general, the variation in total emissions among different seasons was small, which is consistent with other studies (Corbett et al., 1999; Fan et al., 2016). Therefore, meteorological differences were the dominant factor affecting the seasonal differences of ship-related impacts on air quality. Therefore, we've modified this sentence into "higher shipping impacts were expected in June because prevailing winds from the summer monsoon are directed from the ocean to the shore, along with higher ship emissions in summer".

**Revisions in the manuscript:**
*1. Page 7, line 1-3: "Two contrasting months in the year 2015, January and June, were selected to compare the seasonal effects. Higher shipping impacts were expected in June because prevailing winds from the summer monsoon are directed from the ocean to the shore."*

8. Page 6, lines 1-2. I would like to see some discussion on the limitations of AIS in this. It is not used by all ships listed by the authors. The way the text is written now implies that all the ship classes listed here is covered by AIS, which is not necessarily the case since inland traffic may be incompletely represented.

**Response:**
Thank you for the suggestion. The way the text was written was not fully appropriate and we've modified the sentence into "AIS data includes international ships, coastal ships, and inland-water ships, but some river ships could be not covered by AIS data". In addition, we have added section 3.4 Limitations to address the limitations of AIS data as a part of it.

**Revisions in the manuscript:**

1. *Page 7, line 9-10: "AIS data includes international ships, coastal ships, and inland-water ships, but some river ships could be not covered by AIS data."*

2. *Page 19, line 14-29; page 20, line 1-10:*

*"**3.4 Limitations***

*Limitations in the study were mainly related to some missing information, assumptions and model inputs during estimation of shipping emissions. When estimating shipping emission inventory, underestimations of actual emissions may be introduced by missing information. For example, AIS data has a high coverage of coastal vessels, but many inland vessels are not equipped with AIS. Therefore, emissions from those inland vessels without AIS devices were supplemented by using 2015 vessel call data provided by Shanghai MSA and Shanghai Municipal MSA. However, emissions from fishing boats were probably underestimated because AIS devices on some fishing boats may not be in use. Similarly, limited information exists on auxiliary boilers in the Lloyd's register and CCS databases so we calculated the main engine and auxiliary engine emissions but did not consider auxiliary boiler emissions in this study, which may cause underestimation of shipping emissions.*

*We did not consider the external effects of water flow, wind, and waves when calculating engine power for ships going over the region. These factors may increase fuel consumption of individual vessels by as much as 10% to 20%, while the effects of waves on emissions estimations over extensive geographical regions are negligible (Jalkanen et al., 2009; Jalkanen et al., 2012). The downstream of the Yangtze River is located in the geographically plateau region, and the river flow is below 0.5 m/s (Song and Tian, 1997; Xue et al., 2004). For Shanghai, located at the end of mouth of the Yangtze River to the East China Sea with a flat terrain, the river flow is very slow. Given that ships traveling the Yangtze River near Shanghai have speeds over ground (SOG) of about 5-10 knots (3-5 m/s), the relative ratios of water flow to the SOG is within 20%. This would introduce some uncertainties. In our future work, we will fill the gap in the basic ship data and consider the external effects when building the shipping emission inventory."*

9. Page 6, lines 4-6. Does the material obtained from MSA include boats? I would imagine that boats outnumber ships by at least an order of magnitude. Was any consideration given to boat contributions to emissions? Boats may not be the biggest source of CO2, NOx or SOx, but they are a significant source of VOCs and CO.

**Response:**

Thank you for the question. The material, 2015 vessel call data, obtained from MSA includes information on some registered inland boats. However, emissions from fishing boats were probably underestimated, since some AIS devices on fishing boats may not be in use. Discussion on underestimation of emissions from fishing boats has been added to the new section on limitations.

**Revisions in the manuscript:**

1. *Page 19, line 16-25: "When estimating shipping emission inventory,*

*underestimations of actual emissions may be introduced by missing information. For example, AIS data has a high coverage of coastal vessels, but many inland vessels are not equipped with AIS. Therefore, emissions from those inland vessels without AIS devices were supplemented by using 2015 vessel call data provided by Shanghai MSA and Shanghai Municipal MSA. However, emissions from fishing boats were probably underestimated because AIS devices on some fishing boats may not be in use. Similarly, limited information exists on auxiliary boilers in the Lloyd's register and CCS databases so we calculated the main engine and auxiliary engine emissions but did not consider auxiliary boiler emissions in this study, which may cause underestimation of shipping emissions.*"

10. Page 6, lines 7-12. Use of speed entries of AIS. How did you count for the water flow? You have concentrated the study on an area which is along a large river, which means that there is a significant water flow. When a case like this occurs, speed over water is not the same as the speed over ground indicated by the AIS. If power predictions are based on speed over ground, then power prediction will fail. Have the authors considered this aspect?

**Response:**

Thank you for pointing out the potential importance of water flow. The Yangtze River is indeed a large river with the length of 6300 km, divided into the upstream, middle stream and the downstream. The average river flow rate of the upstream and middle stream is in the range of 0.5-4.5 m/s due to the great height difference (Li, 2016; Xue et al., 2004). But for the downstream region, the geographically plateau region, the river flow is below 0.5 m/s (Song and Tian, 1997; Xue et al., 2004). For Shanghai, located at the end of the Yangtze River with a flat terrain, the river flow is very slow. Given that ships traveling the Yangtze River near Shanghai have speeds over ground (SOG) of about 5-10 knots (3-5 m/s), the relative ratios of water flow to the SOG is within 20%. We did not consider the influence of the water flow when calculating engine power for ships in this area for now. But your suggestions will be very useful for our further work extended for larger domain covering the middle and upstream Yangtze River. Also, we have added some discussion of water flow to the new section on limitations in Section 3.4.

**Revisions in the manuscript:**

*1. Page 19, line 26-29; page 20, line 1-7: "We did not consider the external effects of water flow, wind, and waves when calculating engine power for ships going over the region. These factors may increase fuel consumption of individual vessels by as much as 10% to 20%, while the effects of waves on emissions estimations over extensive geographical regions are negligible (Jalkanen et al., 2009; Jalkanen et al., 2012). The downstream of the Yangtze River is located in the geographically plateau region, and the river flow is below 0.5 m/s (Song and Tian, 1997; Xue et al., 2004). For Shanghai, located at the end of mouth of the Yangtze River to the East China Sea with a flat terrain, the river flow is very slow. Given that ships traveling the Yangtze River near Shanghai have speeds over ground (SOG) of about 5-10 knots (3-5 m/s), the relative ratios of water flow to the SOG is within 20%. This would introduce some*

*uncertainties. In our future work, we will fill the gap in the basic ship data and consider the external effects when building the shipping emission inventory."*

11. Page 6, lines 13-14. This is a rather drastic assumption. Have you thought about linking the fuel type or S content to engine specifications? There are technical reasons why some engines cannot use certain types of fuels, but have authors chosen to neglect these limitations completely?

**Response:**

Thank you for the question. Sulfur content is related to fuel type, and some engines can only use certain fuel types. In this study, the fuel type and sulfur content were linked to engine specifications in the model to estimate shipping emission (Fan et al., 2016). We have clarified the link among sulfur content, fuel type and engine type in the supporting information.

**Revisions in the manuscript:**

*1. Page 7, line 22-24: "Assumptions regarding the fuel types, sulfur contents and engine types, and sources of emission factors, low load adjustment multipliers, and control factors are provided in section S.2 of the supporting information."*

*2. Supporting information, Page 3, line11-25: "The two most common fuel oils used in ships are residual oil (RO) and marine distillates (MD). In general, RO is used in the main engine, and the fuel sulfur content is approximately 2.7%, MD is used in the auxiliary engine, and the sulfur content is approximately 0.5%. On the basis of data on ships passing by the Port of Shanghai provided by the largest Chinese heavy fuel oil (HFO) supplier, China Marine Bunker (CMB), the sulfur content of the fuel used by the main engines in domestic vessels ranges from 0.2% to 2.0%, and the sulfur content of the fuel used by the main engines in ocean-going vessels ranges from 1.9% to 3.5%. In this study, we adjusted the sulfur content of the fuel used by the main engines in domestic vessels to 1.5% and that of ocean-going vessels to 2.7%. The amount of $SO_2$ emitted is directly affected by the sulfur content of the fuel; therefore, when main engine emissions were estimated by the model, the emissions of domestic vessels were amended correspondingly. The main engine category was sorted into slow speed diesel (SSD), medium speed diesel (MSD), and high speed diesel (HSD) based on the engine revolutions per minute (RPM), and the largest auxiliary engine category was MSD. The type of engine was judged first according to the RPM of the main engine in Lloyd's database. The emission factors of the different types of engines differ considerably."*

12. Page 7, lines 3-6. "Shopping" of emission data piece by piece from various data providers may lead to unexpected side effects, which can arise from the fundamental assumptions used in emission inventory construction work. The CO and VOC emissions both result from incomplete combustion of fuel and there is a high probability that these two are linked. Did the authors check what the CO/VOC share in IIASA inventory was and how different the CO/VOC share was in the combined inventory?

**Response:**

Thank you for the question. We've checked the CO/VOC share in both IIASA inventory and the combined inventory. For the national scale land-based emission inventory, the emissions of CO and VOC in IIASA inventory are $1.84 \times 10^5$ kt/yr and $2.39 \times 10^4$ kt/yr, respectively, and the CO/VOC share is 7.7. The emissions of CO and VOC in the combined inventory are $1.79 \times 10^5$ kt/yr and $2.37 \times 10^4$ kt/yr, respectively, and the CO/VOC share is 7.5. Therefore, the CO/VOC share in the combined inventory is very close to the one in IIASA inventory. Besides, the CO emission from IIASA inventory was only used for national-scale nested domain in the air quality modelling (domain 1: the whole China, and domain 2: East China). For local-scale domain in the air quality modelling (domain 3: the YRD, and domain 4: Shanghai city), Shanghai Academy of Environmental Science (SAES) provided the complete land-based emission database.

**Revisions in the manuscript:**
*1. Page 8, line 19-25: "Building the national-scale land-based emission inventory by merging data from two datasets may introduce uncertainties. In case of the large uncertainty, the ratio of CO to VOC was checked in this study. CO and VOC emissions both result from incomplete combustion of fuel and are likely to be related (von Schneidemesser et al., 2010; Wang et al., 2014). The ratio of CO to VOC was 7.7 in the IIASA inventory and 7.5 in the final combined inventory. Thus, the CO/VOC shares in these two inventories were very close and the use of the final combined inventory is acceptable."*

13. Page 7, lines 16-21. What was the temporal resolution of the ship emission inventories used in this work?
**Response:**
Thank you for the question. The monthly ship emission inventories were used in this work. We've clarified this in the revised manuscript.
**Revisions in the manuscript:**
*1. Page 7, line 12-14: "Emissions from ships entering the geographic domains for YRD or Shanghai were calculated using the AIS-based model developed by Fan et al. (Fan et al., 2016), and monthly shipping emissions for January and June were used in the air quality model to capture the seasonal variation to expect more accurately than annual shipping emissions with no monthly variations."*

14. Page 8, lines 15-17, the last sentence. There is no uncertainty involved in atmospheric measurements? Really? These can be tens of percent, easily. Cross comparisons of AQ measurement results between instruments can deviate significantly, depending on the equipment used.
**Response:**
Thank you for the question. Indeed, uncertainties could exist in the measurement data, and we have revised this sentence.
**Revisions in the manuscript:**
*1. Page 10, line 11-16: "The deviations between the simulation results and the*

*monitoring data were mainly due to the uncertainties of emission inventories and some deficiencies of meteorological and air quality models. However, there were also uncertainties associated with the measurements themselves and the comparison of grid-based predictions to measurements at point locations."*

15. Page 8, lines 25-29. I agree that population weighted approach has some merit, but that still is an incomplete representation of human activity. The approach used here assumes that people spend all their time at home and do not consider realistic behavior of people. There are some studies that take this into account (see for example Soares et al, GMD, 2014).

**Response:**

Thank you for the question. The reviewer is correct that population-weighted exposures are an approximation of exposure given the more complex reality of where people spend their time.

Soares et al. represent a class of methods that have been in the literature for many years but that have rarely been applied on a large population scale given the intensity of data requirements. The underlying concern is that misclassifying individuals' exposure may introduce bias or reduced precision in ultimate estimates of the population impacts on health. However, this issue of exposure misclassification has been carefully studied in epidemiological studies, including those of air pollution, where reliance on broad geographic characterizations of exposure is common. In general, the findings from epidemiology suggest that, in theory and in practice, the use of these population exposures estimate likely leads to random error in in the true exposures of individuals and has the effect of dampening the observed effect estimates – that is that they are biased low. There is some evidence that when exposure estimates better approximate personal exposures, that the size of the effect estimates increases. The bottom line is that the large population-based epidemiological studies that form the basis for our understanding of air pollution health effects have not relied on methods like those in Soares et al. Population-weighted exposures have been adopted as the basis for estimating the burden of disease from air pollution in the Global Burden of Disease project run by the Institute for Health Metrics and Evaluation (Cohen et al., 2017). IHME's methodology is also now used by the World Health Organization.

**Revisions in the manuscript:**

*1. Page 11, line 1-8: "Soares et al. (2014) built a refined model for evaluating population exposures to ambient air pollution in different microenvironment. In the absence of detailed individual exposure estimates, population-weighted $PM_{2.5}$ concentrations are a better approximation of potential human exposure because they give proportionately greater weight to concentrations in areas where most people live. Population-weighted exposures have been adopted as the basis for estimating the burden of disease from air pollution in the Global Burden of Disease project run by the Institute for Health Metrics and Evaluation (Cohen et al. 2017). IHME's exposure*

*methodology is also now used by the World Health Organization."*

16. Page 9, lines 1-2. It seems that the annual estimate is based on two months of actual data. Why not using data for the whole year? This would remove one source of uncertainty from the final results. The lines 9-10 seem to suggest that data for the whole year 2015 was available for the authors.

**Response:**

Thank you for the question. Due to the limitation of getting the national-scale AIS data for the whole year from the marine-time department, only data in some representative month like January and June are available for our study. Therefore, we used the average values of these two months to estimate annual shipping emissions in whole China. But we have full-year AIS data in Yangtze River Delta (YRD), and the estimates of annual shipping emissions in YRD scale and Shanghai city scale in the manuscript were based on the full-year data. We have clarified the data limitations for national shipping emission estimate in the revised manuscript.

**Revisions in the manuscript:**

1. *Page 11, line 12-14:* "*Due to limitation of the data source, the national-scale AIS data in this study only covered the representative months of January and June 2015, while the YRD-scale AIS data covered 2015 full year.*"

17. Page 9, lines 18-20. The largest contribution to emissions comes from sources close to the shore. This underlines the importance of including all waterborne traffic sources and consideration of water flow/speed issue. Some discussion of these topics should be included in the manuscript.

**Response:**

Thank you for this comment. As we responded to "comment 8" and "comment 10", we've added some discussion on the limitation of AIS data and neglecting water flow.

**Revisions in the manuscript:**

1. *Page 19, line 16-25:* "*When estimating shipping emission inventory, underestimations of actual emissions may be introduced by missing information. For example, AIS data has a high coverage of coastal vessels, but many inland vessels are not equipped with AIS. Therefore, emissions from those inland vessels without AIS devices were supplemented by using 2015 vessel call data provided by Shanghai MSA and Shanghai Municipal MSA. However, emissions from fishing boats were probably still underestimated because most AIS devices on fishing boats were not in use.*"

2. *Page 19, line 26-29; page 20, line 1-7:* "*We did not consider the external effects of water flow, wind, and waves when calculating engine power for ships going over the region. These factors may increase fuel consumption of individual vessels by as much as 10% to 20%, while the effects of waves on emissions estimations over extensive geographical regions are negligible (Jalkanen et al., 2009; Jalkanen et al., 2012). The downstream of the Yangtze River is located in the geographically plateau region, and the river flow is below 0.5 m/s (Song and Tian, 1997; Xue et al., 2004). For Shanghai, located at the end of mouth of the Yangtze River to the East China Sea with*

*a flat terrain, the river flow is very slow. Given that ships traveling the Yangtze River near Shanghai have speeds over ground (SOG) of about 5-10 knots (3-5 m/s), the relative ratios of water flow to the SOG is within 20%. This would introduce some uncertainties. In our future work, we will fill the gap in the basic ship data and consider the external effects when building the shipping emission inventory."*

18. Page 10, lines 8-10. Authors identify ships as a significant source of VOCs. Have you considered the role of small boats in VOC emissions? The VOC emission levels allowed for boat engines are significantly higher than those of marine diesel engines and there are a lot of small engines in boats.

**Response:**

Thank you for the question. "The emissions of $SO_2$, $NO_X$, and $PM_{2.5}$ from inland-water ships and coastal ships accounted for the majority of primary emissions from all shipping related sources in Shanghai port, ranging from 72% for VOCs to about 99% for $SO_2$." Here the proportion is relative to all shipping related sources, which include inland-water ships, coastal ships, container-cargo trucks and port terminal equipment. In this study, the percentage of VOC emissions from ships relative to all pollution sources was not significant, which was 0.3% in YRD region and 0.6% in Shanghai. We have clarified the text and added some discussion on the underestimation of emissions from small boats to the limitations section.

**Revisions in the manuscript:**

*1. Page 11, line 20-23: "Based on the whole year 2015 AIS data, the annual emissions of $SO_2$, $NO_X$, $PM_{2.5}$, and $VOC_s$ from shipping sectors in YRD region were estimated at $2.2 \times 10^5$ tons (one third of the value for China), $4.7 \times 10^5$ tons, $2.7 \times 10^4$ tons, and $1.2 \times 10^4$ tons, respectively, which accounted for 7.4%, 11.7%, 1.3%, and 0.3% of the total emissions from all sources in the YRD in 2015."*

*2. Page 12, line 22-26: "The emissions of $SO_2$, $NO_X$, $PM_{2.5}$, and $VOC_s$ from inland-water ships and coastal ships accounted for the majority of primary emissions from all shipping related sources in Shanghai port, ranging from 72% for VOCs to about 99% for $SO_2$. They comprised about 17.4% of $SO_2$, 24.5% of $NO_x$, 5.2% of $PM_{2.5}$ and 0.6% of $VOC_s$ emissions from all pollution sources in Shanghai."*

*3. Page 19, line 16-25: "When estimating shipping emission inventory, underestimations of actual emissions may be introduced by missing information. For example, AIS data has a high coverage of coastal vessels, but many inland vessels are not equipped with AIS. Therefore, emissions from those inland vessels without AIS devices were supplemented by using 2015 vessel call data provided by Shanghai MSA and Shanghai Municipal MSA. However, emissions from fishing boats were probably underestimated because AIS devices on some fishing boats may not be in use. Similarly, limited information exists on auxiliary boilers in the Lloyd's register and CCS databases so we calculated the main engine and auxiliary engine emissions but did not consider auxiliary boiler emissions in this study, which may cause underestimation of shipping emissions."*

19. Page 12, lines 19-21. If the atmospheric conversion of gaseous SO2 to particulate

SO4 takes about a week (reacts with OH), why is the 12 nm distance relevant in this aspect? Surely during one week the gaseous SO2 travels further that 12 nm during that time and it cannot be used as an only explanation why ships further out than 12 nm do not contribute to SO2.

**Response:**

Thank you for the question. The results showed that shipping within 12 NM was a major contributor to ship-related $SO_2$ concentrations in core YRD cities, which accounted for at least 78% of the ship-related contribution. Shipping further out than 12 NM accounted for 2% to 17% of the ship-related contribution to $SO_2$ concentrations in different core YRD cities. Here 12 NM is a reference distance for comparison of different transport range between precursors like $SO_2$ and aerosol. Shipping within 12 NM was dominant in ship-related $PM_{2.5}$ concentrations in core YRD cities. However, the contribution from shipping further out 12 NM to $PM_{2.5}$ also substantial, accounting for 17% to 49% of the ship-related contribution (especially busy north-south shipping lanes 24-96 NM, accounting for 12 to 39%). It indicates that the ship-related $PM_{2.5}$ concentrations could also be substantially affected by shipping beyond 12 NM, especially when compared with the $SO_2$ result. That also implied that the future ECA boundary should consider multiple air pollutants synchronically. This comparison of results could have policy implication and has been clarified in the revised manuscript. In addition, we've expanded the possible reasons why ships further out than 12 NM had much smaller impact on land $SO_2$ concentrations.

**Revisions in the manuscript:**

*1. Page 15, line 13-25: "Shipping emissions beyond 12 NM had limited contribution to $SO_2$ concentrations in 16 core YRD cities, implying that the boundary of 12 NM might be suitable for regulating $SO_2$ emissions. This could also be proved by Schembari et al., (2012), who reported that statistically significant reductions of $SO_2$ levels (66% to 75%) were found in 3 out of the 4 European harbours, 5 months after the implementation of the EU directive 2005/33/EC that requires all ships at berth or anchorage in European harbours use fuels with a sulfur content of less than 0.1% from January 2010. The quicker chemical reaction and shorter lifetime of $SO_2$ may explain why ships further out than 12 NM had much smaller impact on land ambient $SO_2$ concentrations (Collins et al., 2009; Krotkov et al., 2016). $SO_2$ reacts under tropospheric conditions via both gas-phase processes (with OH) and aqueous-phase processes (with $O_3$ or $H_2O_2$) to form sulfate aerosols, and is also removed physically via dry and wet deposition (Seinfeld and Pandis, 2006). The sulfur deposition due to shipping emissions is mainly contributed by the dry depositions (Chen et al.,2019). In the Planet boundary layer (PBL), $SO_2$ has short lifetimes (less than 1 day during the warm season) and are concentrated near their emission sources (Krotkov et al., 2016)."*

*2. Page 16, line 8-11: "The results of these YRD analyses suggest that although ambient $SO_2$ concentrations were mainly affected by shipping inland or within 12 NM, expanding China's current DECA to around 100 NM or more would reduce the majority of the impacts of shipping on regional $PM_{2.5}$ pollution."*

20. Figure 1. The legend text font size should be increased, it is very small reading as it is now. It is especially tough to read the text of the right hand side zoomed images.

**Response:**

Thank you for the suggestion. The text font size has been increased.

**Revisions in the manuscript:**

*Figure 15*

[Figure]

*Figure 1. Geographic location of the study area YRD/Shanghai with population density in 2015. 16 core cities in YRD and 16 administrative districts in Shanghai are noted on the map. The smaller administrative districts are labeled with numbers: Putuo (1), Jingan (2), Hongkou (3), Yangpu (4), Huangpu (5), Changning (6), Xuhui (7).*

21. Figure 2. This figure is confusing. If the symbols represent measurement values, I cannot see any numerical values linked to the symbols. If the colors correspond to gridded model concentrations, that is fine, but the measured values cannot be determined from these images. Perhaps another form of graphic could be used to provide the comparisons?

**Response:**

Thank you for this comment. The colors correspond to both gridded model concentrations and measured values (in circle). We have increased the circle size so that the fill color is now more visible. This figure has been moved to supporting information (Figure S2).

**Revisions in the manuscript:**

*Supporting information, Figure S2*

[Figure]

*Figure S2. The simulated (grid) and observed (circles) SO$_2$ concentration distribution in YRD region, in January 2015 (a) and June 2015 (c); the simulated (grid) and observed (circles) PM$_{2.5}$ concentration distribution in YRD region, in January 2015 (b) and June 2015 (d)*

22. Figure 3. The legend texts are very small in this figure, too.

**Response:**

Thank you for the suggestion. The legend text font size has been increased.

**Revisions in the manuscript:**

*Figure 2*

[Figure]

*Figure 2. SO₂ emissions in 2015 from (a) shipping traffic in China (the average value of January and June) at resolution of 81km ✕ 81km; (b) ships in different offshore coastal areas (inland-water and within 12 NM, 12-24 NM, 24-48 NM, 48-96 NM and 96-200 NM) in the YRD region, at resolution of 9km ✕ 9km; (c) inland-water ships and coastal ships in Shanghai, at resolution of 1km ✕ 1km; and (d) container-cargo trucks and port terminal equipment in Shanghai, at resolution of 1km ✕ 1km. The black line in (c) refers to the division line between the inland water and coastal water for Megacity Shanghai defined in this study.*

23. Figure 5, I would welcome some discussion why the distance to the shore is relevant in this context. Are the authors trying to see whether it is useful to limit the distance of regulated emissions to a specific value or what is the reasoning of choosing these distance bins?

**Response:**

Thank you for the question. The distance boundary of current Domestic Emission Control Areas (DECA) in China is 12 NM zone along the coastline. We are assessing the impacts of shipping within 12 NM as well as shipping in offshore coastal areas beyond 12 NM. We referred to ICCT's working paper (Mao et al., 2017) to choose

the distance bins (12 NM, 24 NM, 48 NM, 96NM, 200NM) in this study. Among these distance bins, 12 NM is the boundary of current DECA in China and 200 NM is the boundary of ECA designated by IMO. In addition, we considered that shipping at further distances could have a smaller impact on air quality on the land, therefore, the values between 12 NM and 200 NM were doubled in order to make a better comparison among these offshore coastal areas. The results can provide evidence when a specific value is considered for the distance of regulated emissions in future ECA policy in China. We've clarified the reason of choosing the distance bins in the revised manuscript.

**Revisions in the manuscript:**

*1. Page 5, line 30; page 6, 1-2: "The results of this study could be informative to the consideration of the distance of regulated emissions in the design of future emissions control areas for shipping in YRD"*

*2. Page 6, line 9-11: "Then, we used WRF-CMAQ model to evaluate the impacts on air quality from shipping emissions in different offshore coastal areas (within 12 NM including inland waters, 12-24 NM, 24-48 NM, 48-96 NM, and 96-200 NM) in the YRD region. We referred to ICCT's working paper (Mao et al., 2017) to choose the distance bins between 12 NM (the boundary of current China's DECA) and 200 NM (the boundary of ECA designated by IMO) in this study."*

24. Figure 9. Texts are too small, especially in the two top images.

**Response:**

Thank you for the suggestion. The text font size has been increased.

**Revisions in the manuscript:**

*Figure 8*

[Figure]

*Figure 8.* *Population-weighted $PM_{2.5}$ and average $PM_{2.5}$ caused by different ship-related sources in Shanghai, in January(a) and in June (b); population-weighted $PM_{2.5}$ caused by all pollution sources (c, g), inland-water ships (d, h), coastal ships (e, i) and container-cargo trucks and port terminal equipment (f, j) in 16 districts in Shanghai, in January 2015(c-f) and June 2015 (g-j)*

25. Table 1. Are these daily, monthly or annual values? There is no indication of the timeline here? This data does not tell me very much of how well the model is able to capture the temporal variability of pollution peaks. Could a line graph be used here instead? This would help to see how well the model is able to capture the air concentrations.

**Response:**

Thank you for this comment. Observed data (Obs.) and simulated data (Sim.) for each city are the average of monthly values of January and June case. The statistical metrics of NMB, NME, RMSE and *r* were calculated based on the daily-average observed and simulated data. We've clarified the timeline in the caption of Table 1.

We've made line graphs of temporal variability. However, if we put all the line graphs in the manuscript, it would be too many (32 line graphs for 16 cities and two pollutants). So we chose line graphs of four representative cities (two coastal cities and two inland cites) to be presented in the supporting information.

**Revisions in the manuscript:**

*1. Table 1, caption*

*"Table 1 Statistical metrics of the model evaluation. Observed data (Obs.) and*

*simulated data (Sim.) for each city are the average of monthly values of January and June case. NMB, NME, RMSE and r were calculated based on the daily-average observed and simulated data."*

2. Supporting information, Figure S3

[Figure]

*Figure S3 Daily variability of simulated (sim.) and observed (obs.) SO$_2$ concentrations (a, c, e, g) and PM$_{2.5}$ concentrations (b, d, f, h) in four representative cities, including two coastal cities – Shanghai (a, b) and Ningbo (c, d), and two inland cites – Hangzhou (e, f) and Suzhou (g, h).*

26. Table 2. No units are given?

**Response:**

Thank you for the question. The units were given in the left column which may not be very obvious. We've clarified the units in the title.

**Revisions in the manuscript:**

*Table 2. Primary emissions (ton/yr), emission share in all shipping emission (%) and emissions density (ton/yr/km$^2$) from shipping at different boundaries in YRD region[a] in 2015*

| | Pollutants | Within 12 NM | 12-24 NM | 24-48 NM | 48-96 NM | 96-200 NM |
|---|---|---|---|---|---|---|
| *Shipping* | $SO_2$ | $1.3 \times 10^5$ | $1.4 \times 10^4$ | $2.5 \times 10^4$ | $3.2 \times 10^4$ | $1.3 \times 10^4$ |
| *emission* | $NO_x$ | $3.6 \times 10^5$ | $2.0 \times 10^4$ | $3.5 \times 10^4$ | $4.5 \times 10^4$ | $1.8 \times 10^4$ |
| *inventory* | $PM_{2.5}$ | $1.3 \times 10^4$ | $2.4 \times 10^3$ | $4.5 \times 10^3$ | $5.4 \times 10^3$ | $1.5 \times 10^3$ |
| *(ton/yr)* | $VOC_s$ | $7.9 \times 10^3$ | $8.3 \times 10^2$ | $1.3 \times 10^3$ | $1.5 \times 10^3$ | $3.0 \times 10^2$ |
| *Emission* | $SO_2$ | 61.4 | 6.4 | 11.4 | 14.9 | 5.8 |
| *share in all* | $NO_x$ | 75.0 | 4.1 | 7.4 | 9.6 | 3.9 |
| *shipping* | $PM_{2.5}$ | 48.4 | 9.0 | 16.9 | 20.2 | 5.5 |
| *emission (%)* | $VOC_s$ | 66.6 | 7.0 | 11.2 | 12.6 | 2.6 |
| *Emission* | $SO_2$ | 0.66 | 0.54 | 0.49 | 0.33 | 0.06 |
| *density* | $NO_x$ | 1.74 | 0.86 | 0.77 | 0.51 | 0.08 |
| *(ton/yr/km$^2$)* | $PM_{2.5}$ | 0.08 | 0.06 | 0.06 | 0.04 | 0.01 |
| | $VOC$ | 0.05 | 0.02 | 0.01 | 0.01 | 0.001 |

*a. domain 3*

27. Table 3. No units given?

**Response:**

Thank you for the question. The units were given in the left column which may not be very obvious. We've clarified the units in the title.

**Revisions in the manuscript:**

*Table 3. Primary emissions (ton/yr) and emission share in all pollution sources (%) from individual ship-related source in Shanghai[a] in 2015*

| | Ship-related source | $SO_2$ | $NO_x$ | $PM_{2.5}$ | VOC |
|---|---|---|---|---|---|
| *Emission inventory (ton/yr)* | *Inland-water ships[b]* | $3.3 \times 10^4$ | $9.2 \times 10^4$ | $0.40 \times 10^4$ | $0.27 \times 10^4$ |
| | *Coastal ships[c]* | $1.6 \times 10^4$ | $2.9 \times 10^4$ | $0.18 \times 10^4$ | $0.067 \times 10^4$ |
| | *Container-cargo trucks* | 0.0 | $1.8 \times 10^4$ | $0.064 \times 10^4$ | $0.11 \times 10^4$ |
| | *Port terminal equipment[d]* | $0.0021 \times 10^4$ | $0.18 \times 10^4$ | $0.0057 \times 10^4$ | $0.022 \times 10^4$ |
| *Emission share in all pollution sources in Shanghai (%)* | *Inland-water ships* | 11.8 | 18.7 | 3.6 | 0.5 |
| | *Coastal ships* | 5.6 | 5.8 | 1.6 | 0.1 |
| | *Container-cargo trucks* | 0.0 | 3.7 | 0.6 | 0.2 |
| | *Port terminal equipment* | 0.01 | 0.36 | 0.05 | 0.04 |

*a. domain 4*

*b. defined as ships operate in both the outer port and in the inner river region of Shanghai Port, which include Yangtze River, Huangpu River and other river ways in Shanghai*

*c. includes China coastal and international ships*

*d. includes cranes and forklifts used for internal transport*

Supplementary material, S1

28. The authors seem to apply the Starcrest methodology in their emission modeling. Page 2, text under Eq (1). Maximum speed and design speed of ships are two different things and IHS data often mentions economic speed. Which was one was actually used in the analysis?

**Response:**

Thank you for the question. Maximum speed was used in the analysis. We've corrected this sentence into "where ActSpeed is the actual speed when ship is cruising and MaxSpeed is the maximum speed for the ship".

**Revisions in the manuscript:**

*1. Supporting information, Page 2, line 8-9: "where ActSpeed is the actual speed when ship is cruising and MaxSpeed is the maximum speed for the ship"*

29. Page 2, near Eq (4), aux boiler use. Did the authors consider the exhaust boilers at all in this regard? Also, the installed boiler capacity is difficult to determine from ship databases, because this field is not provided. I would like to know where the installed boiler data comes from.

**Response:**

Thank you for the question. We did not consider the exhaust boilers because limited information on boilers could be found in Lloyd's database and China Classification

Society (CCS) database. We've add some discussion about the underestimation of boiler emissions in the new section on limitations.

**Revisions in the manuscript:**

*1. Supporting information, page 2, line 28-30: "However, auxiliary boiler emissions were not considered in this study because limited auxiliary boiler information exists in the Lloyd's register and Chinese Classification Society (CCS) database."*

*2. Page 19, line 22-25: "Similarly, limited information exists on auxiliary boilers in the Lloyd's register and CCS databases so we calculated the main engine and auxiliary engine emissions but did not consider auxiliary boiler emissions in this study, which may cause underestimation of shipping emissions."*

30. Page 2, last paragraph. Authors make reference to Lloyds, 2009 which is not listed in the bibliography provided for S1. Also, why refer to data from 2009 if the AIS data is for 2015. How were the ships built during 2009-2015 treated?

**Response:**

Thank you for the question. In fact, the 2015 Lloyd's register was used in this study. We have corrected this mistake and also listed the reference in the bibliography for S1.

**Revisions in the manuscript:**

*1. Supporting information, page 2, line 35-36: "For ships available in Lloyd's register (now IHS-Fairplay) (Lloyd's register, 2015)"*

*2. Supporting information, References: "Lloyd's register (IHS Fairplay). 2015"*

31. Page 2, last paragraph. Authors assume all inland waterway vessels to have 7000 kW engine? No effort was made to identify these vessels and use proper description of installed power?

**Response:**

Thank you for the question. We did not assume all inland waterway vessels to have 7000 kW engine. Information of some domestic ships is available in Chinese Classification Society (CCS) database, and description of installed power was used according to the information in the database. But for those domestic ships which are not available in Lloyd's register and CCS database, their main engine power was assumed to be 7000 kw by default, which was close to the domestic ships from the CCS database (with main engine power mainly ranging from 4000 kw to 6000 kw), and below the East China Sea-going ships in Lloyd's register (with main engine power mainly ranging from 11000 kw to 14000 kw). We've clarified this in the revised manuscript. Also we will fill the gap in the basic ship data in future work.

**Revisions in the manuscript:**

*1. Supporting information, page 3, line 2-6: "Information of some domestic ships is available in CCS database, but for those ships unavailable in the database, the main engine power was assumed to be 7000 kw by default, which was close to the domestic ships from the CCS database (with main engine power mainly ranging from 4000 kw to 6000 kw) and below the East China Sea-going ships in Lloyd's register (with main*

*engine power mainly ranging from 11000 kw to 14000 kw)"*

32. Page 3, S2, first paragraph. "Table S1 lists emission factors used in the present study". This is not true and the emission factor table is missing.

**Response:**

Thank you for the correction. Since the emission factor table was already in our previous work (Fan et al. 2016), we did not put the table in the supporting information in this study repeatedly. We are sorry for the written mistake. We've modified this sentence into "Emission factors used in the present study were listed in Fan et al. (2016)."

**Revisions in the manuscript:**

*1. Supporting information, page 3, line 31-32 "Emission factors used in the present study were listed in Fan et al. (2016)."*

*2. Supporting information, References: "Fan, Q., Zhang, Y., Ma, W., Ma, H., Feng, J., Yu, Q., Yang, X., Ng, S. K., Fu, Q., and Chen, L.: Spatial and Seasonal Dynamics of Ship Emissions over the Yangtze River Delta and East China Sea and Their Potential Environmental Influence, Environ. Sci. Technol., 50, 1322-1329, 10.1021/acs.est.5b03965, 2016."*

33. Page 3, second paragraph. Add reference ICF, 2009

**Response:**

Thank you for the correction. The reference has been added.

**Revisions in the manuscript:**

*1. Supporting information, References: "ICF International: Current methodologies in preparing mobile source port-related emission inventories, 2009."*

34. Page 3, second paragraph. OC and EC low load adjustment factors were treated the same way as PM. This is contrary to the behavior of EC and EC as a function of engine load. Authors might want to check the ICCT report "Black Carbon Emissions and Fuel Use in Global Shipping, 2015", Oct 2017 for low load behavior of carbon fraction.

**Response:**

Thank you for this comment. In this study, emission factors were adjusted for loads below 20 % using values from studies conducted in other countries (ICF International, 2009; Starcrest Consulting Group, 2009). Because OC and EC low load adjustment factors were not available in these studies, they were assigned the same as PM. In ICCT's report "Black Carbon Emissions and Fuel Use in Global Shipping, 2015" (ICCT, 2017), the authors mentioned that "Emission factors tend to increase at low loads. Low load adjustment factors from the Third IMO GHG Study 2014 were applied when estimated main engine load fell below 20% for all pollutants except BC,

which is not estimated in the IMO study. In this case, BC EFs are determined from power curves described in the previous section, which already account for changes in BC EFs as a function of engine load." From the power curves of BC EF (shown in the Figure R1 below), it indicated that the BC EFs increase significantly at low loads, especially below 20%. The unit of EF given in ICCT's report (ICCT, 2017) is g/kg fuel, while the unit of EF given in this study is g/kWh, which is hard to make direct comparisons. So we've estimated the proportion of BC emissions in PM emissions (BC/PM). In this study, BC/PM was 0.029, a bit lower than the value 0.045 in ICCT's report. Petzold et al. (2004) measured a BC fraction of 2% of the total particle mass for an engine load of 100%. Erying et al. (2005) estimated shipping emissions based on fuel consumption, and reported 0.05 Tg of BC and 1.67 Tg of $PM_{10}$, and BC/PM was around 0.03. Therefore, the ratio of BC to PM in this study is within a reasonable range. We've added some discussion about the uncertainty brought by the selection of low load adjustment factors in the supporting information.

[Figure]

Figure R1. Black carbon emission factors for 2-stroke engines by fuel type (ICCT, 2017)

**Revisions in the manuscript:**
*1. Supporting information, page 3, line 34-39: "Because adjustment multipliers were not available for organic carbon (OC) and elemental carbon (EC), these pollutants were assigned the same low load adjustment multiplier (LLAM) as PM in the present study, which may introduce uncertainties. In this study, the ratio of BC emissions to PM emissions (BC/PM) was around 2.9%, which falls within the range of 2% to 4.5% in other studies (Comer et al., 2017; Erying et al., 2005; Petzold et al., 2004)."*

**References:**

Aulinger, A., Matthias, V., Zeretzke, M., Bieser, J., Quante, M., and Backes, A.: The

impact of shipping emissions on air pollution in the greater North Sea region - Part

1: Current emissions and concentrations, Atmos. Chem. Phys., 16, 739-758, 10.5194/acp-16-739-2016, 2016.

Cohen, A. J., Brauer, M., Burnett, R., Anderson, H. R., Frostad, J., Estep, K., Balakrishnan, K., Brunekreef, B., Dandona, L., and Dandona, R.: Estimates and 25-year trends of the global burden of disease attributable to ambient air pollution: an analysis of data from the Global Burden of Diseases Study 2015, Lancet, 389, 1907-1918, 2017.

Comer, B., Olmer, N., Mao, X., Roy, B., and Rutherford, D.: Black Carbon Emissions and Fuel Use in Global Shipping, 2015, The International Council on Clean Transportation, 2017. Retrieved from https://www.theicct.org/sites/default/files/publications/Global-Marine-BC-Inventory-2015_ICCT-Report_15122017_vF.pdf

Eyring, V., Köhler, H. W., Aardenne, J., and Lauer, A.: Emissions from international shipping: 1. The last 50 years, J. Geophys. Res., 110, 10.1029/2004jd005619, 2005

Goldsworthy, L., and Goldsworthy, B.: Modelling of ship engine exhaust emissions in ports and extensive coastal waters based on terrestrial AIS data - An Australian case study, 45-60, 10.1016/j.envsoft.2014.09.009, 2015.

ICF International: Current methodologies in preparing mobile source port-related emission inventories, 2009.

Jalkanen, J.-P., Brink, A., Kalli, J., Pettersson, H., Kukkonen, J., and Stipa, T.: A modelling system for the exhaust emissions of marine traffic and its application in the Baltic Sea area, Atmos. Chem. Phys., 9, 9209–9223, doi:10.5194/acp-9-9209-2009, 2009.

Jalkanen, J. P. et al. Extension of an assessment model of ship traffic exhaust emissions for particulate matter and carbon monoxide. Atmos. Chem. Phys. 12, 2641–2659, 2012.

Li W.: Flow test analysis of the Acoustic Doppler Current Profiler in Zhuyu hydrological station on the upper reaches of the Yangtze River, Express Water Resources & Hydropower Information, 37(9), 14-18, 2016. In Chinese.

Mao, X., Cui, H., Roy, B., Olmer, N., Rutherford, D., & Comer, B: Distribution of air

pollution from oceangoing vessels in the Greater Pearl River Delta, 2015, The International Council on Clean Transportation, 2017. Retrieved from http://www.theicct.org/sites/default/files/publications/China-GPRD-Baseline-Emissions-Inventory_ICCT-Working%20Paper_23082017_vF.pdf

Soares, J., Kousa, A., Kukkonen, J., Matilainen, L., Kangas, L., Kauhaniemi, M., Riikonen, K., Jalkanen, J. P., Rasila, T., Hänninen, O., Koskentalo, T., Aarnio, M., Hendriks, C., and Karppinen, A.: Refinement of a model for evaluating the population exposure in an urban area, Geosci. Model Dev., 7, 1855-1872, 10.5194/gmd-7-1855-2014, 2014

Song Z., and Tian C.: Application of the Acoustic Doppler Current Profiler in the Yangtze River Estuary, Hydrology, 6, 31-34, 1997. In Chinese.

Starcrest Consulting Group: Port of Los Angeles Inventory of Air Emissions 2008, Technical Report Revision, 2009.

Petzold, A., Feldpausch, P., Fritzsche, L., Minikin, A., Lauer, P., Kurok, C., and Bauer, H.: Particle emissions from ship engines, J. Aerosol Sci., Abstracts of the European Aerosol Conference, S1095–S1096, 2004

Xue Y., Gu J., and Wei T.: The working principle of the Acoustic Doppler Profiler and its applications in the middle and lower reaches of the Yangtze River, Marine Sciences, 28(10), 24-28, 2004. In Chinese.

---

## Author Comment (AC2) · 9 Mar 2019

**Response to Referee's Comments #2**

Dear Editor and Referees,

We are pleased to submit our responses to all the comments and revision for manuscript acp-2018-1163. We appreciate all the comments and suggestions that are especially helpful. All the referees' comments have been addressed carefully.

Best regards with respect,

Yan Zhang, representing all co-authors

Reviewers' comments are in blue.

Authors' responses are in black.

Revisions in manuscript are in *italic*, underlined.

1. General comments: This study presented the importance of geographical locations of ship emissions to the environmental and human health effects. The manuscript has been well written and organized. Take the YRD region– one of the busiest port cluster in the world as the example, this study result is helpful to understand the meaningful points of future ECA policy. The authors should explicit the key implication through the paper, including the abstract, result and conclusion part.

**Response:**

Thank you for the comments and the suggestions. We have expanded on the key implications of these research results for potential ECA regulations throughout the manuscript.

**Revisions in the manuscript:**

*1. Page 2, line 6-8: "in particular, in the YRD region, expanding the boundary of 12 NM in China's current DECA policy to around 100 NM would include most of the shipping emissions affecting air pollutant exposures, and stricter fuel standards could be considered for the ships on inland rivers and other waterways close to residential regions."*

*2. Page 5, lines 6-15: "In China, a few studies reported the contribution to air pollution from shipping in different offshore coastal areas or individual ship-related sources. For example, Mao et al. (2017) estimated primary emissions from OGVs at different boundaries in the PRD region, and concluded that further expansion of emission control area to 100 NM would provide even greater benefits. However, the impacts of shipping emissions at varying distances from shore on air quality and potential human exposure, which are important when considering ECA policy, have not been rigorously studied. Mao and Rutherford (2018) studied $NO_x$ emissions from three categories of merchant vessels—OGVs, coastal vessels (CVs) and river vessels (RVs) in China's coastal region. But less attention was paid to the impacts of inland waterway traffic and port-related sources like container-cargo trucks and terminal port equipment on air quality and potential human exposure."*

*3. Page 5, line 30; page 6, line 1-3: "The results of this study could be informative to the consideration of the distance of regulated emissions in the design of future emissions control areas for shipping in YRD, or regulations on the sulfur content of fuels for individual ship-related sources in Shanghai."*

*4. Page 16, line 7-10: "The results of these YRD analyses suggest that although ambient ship-related $SO_2$ concentrations were mainly affected by shipping inland or within 12 NM, expanding China's current DECA to around 100 NM or more would reduce the majority of the impacts of shipping on regional $PM_{2.5}$ pollution."*

*5. Page 19, line 10-12: "The results of the analyses of individual shipping-related sources indicated that ship-related sources close to densely-populated areas contribute substantially to population exposures to air pollution."*

*6. Page 20, line 26-29; page 21, line 1-2: "For example, policymakers could consider whether to expand China's current DECA boundary of 12 NM to around 100 NM or more to reduce the majority of shipping impacts on air pollution and exposure or to develop more stringent regulations on the sulfur content of fuels for ships entering inland rivers or other waterways close to residential regions due to their significant influence on local air quality and human exposures in densely populated areas."*

The details should be improved:

2. Page 6-7, 2.2.2 Non-shipping emission inventories part. For the national scale domain and regional scale domain, several sets emission data has been used. The authors should make clearer how they merge the emission together. How did they use 2015 national emission database to make a regional 27 km × 27 km resolution that included 5 pollutants? Did they use spatial interpolation method? Which year are the IIASA data for CO and NH3?

**Response:**

Thank you for the question. The 2015 national emission database (including $PM_{10}$, $PM_{2.5}$, $SO_2$, $NO_x$ and VOCs) at a 27 km × 27 km resolution (Zhao et al., 2018) and 2015 IIASA database at a 0.5 °× 0.5 ° resolution (CO and $NH_3$) (Stohl et al., 2015) was allocated to domain 1 (81 km × 81 km) and domain 2 (27 km × 27 km) by spatial interpolation in Arcgis 10.2. We have clarified the method and the year of IIASA data in the revised manuscript.

**Revisions in the manuscript:**

*1. Page 8, line 17-18:"supplemental emission data on these pollutants in 2015 were obtained from the International Institute for Applied Systems Analysis (IIASA) database (at a 0.5 °× 0.5 ° resolution) (Stohl et al., 2015)."*

*2. Page 8, line 28-29: "National and local emission data were allocated to simulation grids by spatial interpolation in ArcGIS 10.2 (ESRI, 2013)."*

3. Page7, line 15-16: "The initial and boundary conditions for meteorology were generated from the Chinese National Centers for Environmental Prediction (NCEP) Final Analysis (FNL)", here the authors should confirm the NCEP FNL data source.

**Response:**

Thank you for pointing out this. We are sorry for the written mistake. The data source should be "National Centers for Environmental Prediction (NCEP) Final Analysis (FNL)" and it has been corrected in the revised manuscript.

**Revisions in the manuscript:**

*1. Page 9, line 7-8: "The initial and boundary conditions for meteorology were*

*generated from the National Centers for Environmental Prediction (NCEP) Final*
*Analysis (FNL) (NCEP, 2000)"*

4. Page 9, line 12-17: The authors compared the result of YRD shipping emission with Fan et al.'s and Chen et al.'s studies. The authors quoted Liu et al. (2018) to compare the proportion of YRD shipping emissions in whole China. However, Liu et al. (2018) also reported YRD shipping emissions. Why not compare the result with the values in Liu et al. (2018) as well?

**Response:**

Thank you for the suggestion. We've added the comparison with the 2013 YRD shipping emission estimates in Fu et al. (2017). This paper is from the same research group as Liu et al. (2018), but reports more pollutants than in Liu et al. (2018). In addition, the comparison with the results in Fan et al. (2016) has been removed because the values were for the year 2010, much earlier than the baseline year 2015 in this study.

**Revisions in the manuscript:**

*1. Page 11, line 22-25: "The emission estimates of $SO_2$, $NOx$ and $PM_{2.5}$ were slightly lower than Chen et al.'s estimates for 2014 year due to the different temporal or spatial statistical scope (Chen et al., 2019; Fu et al., 2017)."*
*2. References: "Fu, M., Liu, H., Jin, X., and He, K.: National- to port-level* *inventories of shipping emissions in China, Environ. Res. Lett., 12, 114024,* *10.1088/1748-9326/aa897a, 2017."*

5. Page 10, line 12-16: The authors quoted Fu et al. (2012), which used 2010 vessel call data to estimate shipping emissions. I suggest authors reviewed recent studies using AIS data to make comparisons in Shanghai port.

**Response:**

Thank you for the suggestion. We've reviewed the results in Fu et al. (2017) which reported 2013 shipping emissions in Shanghai Port using AIS data. We've added some discussion on the comparison between the values in this study and in Fu et al.'s study.

**Revisions in the manuscript:**

*1. Page 13, line 1-6: "Emissions estimates from this study fall within the range of estimates from other studies (Fu et al., 2012; Fu et al., 2017). On the basis of shipping visa data, Fu et al. (2012) determined that the total amounts of $SO_2$, $NO_X$, and $PM_{2.5}$ in the vicinity of Shanghai port in 2010 were $3.5 \times 10^4$ ton/yr, $4.7 \times 10^4$ ton/yr, and $3.7 \times 10^3$ ton/yr, respectively, substantially lower than estimates in this study. Using AIS data, Fu et al. (2017) reported $5 \times 10^4$ tons of $SO_2$ and $7 \times 10^4$ tons of NOx from shipping in Shanghai port in 2013, close or a bit lower than the results in this study."*

6. Page 12, line 6-15: The contribution to SO2 from ships in different coastal areas was not discussed in this paragraph. But in the following paragraph, the authors discussed cumulative contributions from ships at different distance to both SO2 and PM2.5. It shows no consistency when authors discussed SO2 results throughout the section 3.2.2.

**Response:**

Thank you for this comment. We've supplemented the data of average and peak contribution to $SO_2$ from ships in different coastal areas in Table S4. Also, we've added some discussion in the revised manuscript.

**Revisions in the manuscript:**

*1. Page 14, line 24-25: "The average and peak contributions from the shipping emissions in specific offshore coastal areas to the ambient $SO_2$ and $PM_{2.5}$ concentrations on shore for the two months are listed in Table S4. Shipping emissions beyond 12 NM had a much smaller impact on ambient $SO_2$, which average contributions were below 0.01 μg/m$^3$ and peak contributions were below 0.06 μg/m$^3$ (Table S4)."*

*2. Table S4 Average and peak contributions from ship emissions in different offshore coastal areas to the ambient $SO_2$ and $PM_{2.5}$ concentrations in January and June*

| Offshore distance | Average contribution ($\mu g/m^3$) | | | | Maximum contribution ($\mu g/m^3$) | | | |
|---|---|---|---|---|---|---|---|---|
| | $SO_2$ | | $PM_{2.5}$ | | $SO_2$ | | $PM_{2.5}$ | |
| | January | June | January | June | January | June | January | June |
| Inland and within 12 NM | 0.52 | 0.70 | 0.24 | 0.56 | 6.00 | 8.79 | 1.62 | 4.02 |
| 12-24 NM | 0.005 | 0.007 | 0.01 | 0.04 | 0.03 | 0.05 | 0.05 | 0.20 |
| 24-48 NM | 0.01 | 0.009 | 0.04 | 0.07 | 0.06 | 0.05 | 0.11 | 0.34 |
| 48-96 NM | 0.02 | 0.008 | 0.07 | 0.07 | 0.05 | 0.03 | 0.14 | 0.30 |
| 96-200 NM | 0.00 | 0.001 | 0.003 | 0.01 | 0.004 | 0.003 | 0.02 | 0.05 |

7. Page 14, line 1-6: The authors discussed the population-weighted PM2.5 from both shipping source and all pollution sources. Then, what's the proportion of population-weighted PM2.5 from the shipping source among all pollution sources? I suggest some discussion here.

**Response:**

Thank you for the suggestion. We have added the proportion of population-weighted $PM_{2.5}$ among all pollution sources along with some discussion.

**Revisions in the manuscript:**

*1. Page 17, line 11-15: "Thus, population-weighted $PM_{2.5}$ concentrations from shipping sources accounted for 0.9% to 15.5% of the population-weighted $PM_{2.5}$ concentrations from all pollution sources in June, larger than the contributions of 0.2% to 1.6% in January, which was attribute to higher shipping-related population-weighted $PM_{2.5}$ concentrations in June and higher population-weighted $PM_{2.5}$ concentrations from all pollution sources in January."*

8. Page 15, line 25: The uncertainty analysis is lacked in the section of result and discussion. The uncertainties of shipping emission inventories should be discussed here.

**Response:**

Thank you for pointing out this. We've added section 3.4 Limitations where we

discuss the uncertainties associated with our shipping emission inventories.

**Revisions in the manuscript:**

*1. Page 19, line 14-29; page 20, line 1-10:*

*"3.4 Limitations*

*Limitations in the study were mainly related to some missing information, assumptions and model inputs during estimation of shipping emissions. When estimating shipping emission inventory, underestimations of actual emissions may be introduced by missing information. For example, AIS data has a high coverage of coastal vessels, but many inland vessels are not equipped with AIS. Therefore, emissions from those inland vessels without AIS devices were supplemented by using 2015 vessel call data provided by Shanghai MSA and Shanghai Municipal MSA. However, emissions from fishing boats were probably underestimated because AIS devices on some fishing boats may not be in use. Similarly, limited information exists on auxiliary boilers in the Lloyd's register and CCS databases so we calculated the main engine and auxiliary engine emissions but did not consider auxiliary boiler emissions in this study, which may cause underestimation of shipping emissions.*

*We did not consider the external effects of water flow, wind, and waves when calculating engine power for ships going over the region. These factors may increase fuel consumption of individual vessels by as much as 10% to 20%, while the effects of waves on emissions estimations over extensive geographical regions are negligible (Jalkanen et al., 2009; Jalkanen et al., 2012). The downstream of the Yangtze River is located in the geographically plateau region, and the river flow is below 0.5 m/s (Song and Tian, 1997; Xue et al., 2004). For Shanghai, located at the end of mouth of the Yangtze River to the East China Sea with a flat terrain, the river flow is very slow. Given that ships traveling the Yangtze River near Shanghai have speeds over ground (SOG) of about 5-10 knots (3-5 m/s), the relative ratios of water flow to the SOG is within 20%. This would introduce some uncertainties. In our future work, we will fill the gap in the basic ship data and consider the external effects when building the shipping emission inventory.*

*"*

**Reference:**

Fu, M., Liu, H., Jin, X., and He, K.: National- to port-level inventories of shipping emissions in China, Environ. Res. Lett., 12, 114024, 10.1088/1748-9326/aa897a, 2017.

Liu, H., Meng, Z. H., Shang, Y., Lv, Z. F., Jin, X. X., Fu, M. L., and He, K. B.: Shipping emission forecasts and cost-benefit analysis of China ports and key regions' control, Environ. Pollut., 236, 49-59, 10.1016/j.envpol.2018.01.018, 2018b.

Stohl, A., Aamaas, B., Amann, M., Baker, L. H., Bellouin, N., Berntsen, T. K., Boucher, O., Cherian, R., Collins, W., Daskalakis, N., Dusinska, M., Eckhardt, S., Fuglestvedt, J. S., Harju, M., Heyes, C., Hodnebrog, Ø., Hao, J., Im, U., Kanakidou, M., Klimont, Z., Kupiainen, K., Law, K. S., Lund, M. T., Maas, R., MacIntosh, C. R., Myhre, G., Myriokefalitakis, S., Olivié, D., Quaas, J., Quennehen, B., Raut, J.-C., Rumbold, S. T., Samset, B. H., Schulz, M., Seland, Ø., Shine, K. P., Skeie, R. B., Wang, S., Yttri, K. E., and Zhu, T: Evaluating the climate and air quality impacts of short-lived pollutants, Atmos. Chem. Phys. 15, 10529–10566, https://doi.org/10.5194/acp-15-10529-2015, 2015.

Zhao, B., Zheng, H., Wang, S., Smith, K. R., Lu, X., Aunan, K., Gu, Y., Wang, Y., Ding, D., Xing, J., Fu, X., Yang, X., Liou, K. N., and Hao, J.: Change in household fuels dominates the decrease in PM2.5 exposure and premature mortality in China in 2005-2015, P. Natl. Acad. Sci. USA, 115, 12401-12406, 10.1073/pnas.1812955115, 2018